# DADA: DUAL AVERAGING WITH DISTANCE ADAPTATION

**Mohammad Moshtaghifar**
University of British Columbia
mmoshtaq@cs.ubc.ca

**Anton Rodomanov**
CISPA Helmholtz Center for Information Security
anton.rodomanov@cispa.de

**Daniil Vankov**
Arizona State University
dvankov@asu.edu

**Sebastian U. Stich**
CISPA Helmholtz Center for Information Security
stich@cispa.de

## ABSTRACT

We present a novel universal gradient method for solving convex optimization problems. Our algorithm—Dual Averaging with Distance Adaptation (DADA)—is based on the classical scheme of dual averaging and dynamically adjusts its coefficients based on observed gradients and the distance between iterates and the starting point, eliminating the need for problem-specific parameters. DADA is a universal algorithm whose convergence rate adapts to the local behavior of the objective around its minimizer, through bounds on its local growth. This leads to a single method with explicit, problem-dependent guarantees across a broad range of models, including nonsmooth Lipschitz functions, Lipschitz-smooth functions, Hölder-smooth functions, functions with high-order Lipschitz derivative, quasi-self-concordant functions, and $(L_0, L_1)$-smooth functions. Crucially, DADA is applicable to both unconstrained and constrained problems, even when the domain is unbounded, without requiring prior knowledge of the number of iterations or desired accuracy.

## 1 INTRODUCTION

Gradient methods are among the most popular and efficient algorithms for solving optimization problems arising in machine learning, as they are highly adaptable and scalable across various settings (Bottou et al., 2018). Despite their popularity, these methods face a significant challenge of selecting appropriate hyperparameters, particularly stepsizes, which are critical to the performance of the algorithm. Hyperparameter tuning is one of the standard approaches to address this issue but is a time-consuming and resource-intensive process, especially as models become larger and more complex. Consequently, the cost of training these models has become a significant concern (Sharir et al., 2020; Patterson et al., 2021).

Typically, line-search techniques have been used to select stepsizes for optimization methods, and they are provably efficient for certain function classes, such as Hölder-smooth problems (Nesterov, 2015). However, in recent years, several so-called parameter-free algorithms have been developed which do not utilize line search (Orabona & Tommasi, 2017; Cutkosky & Orabona, 2018; Carmon & Hinder, 2022; Ivgi et al., 2023; Khaled et al., 2023; Mishchenko & Defazio, 2024). Notably, one strategy involves dynamically adjusting stepsizes based on estimates of the initial distance to the optimal solution (Carmon & Hinder, 2022; Ivgi et al., 2023; Khaled et al., 2023). Another approach leverages lower bounds on the initial distance combined with the Dual Averaging (DA) scheme (Defazio & Mishchenko, 2023; Mishchenko & Defazio, 2024). However, these methods primarily focus on nonsmooth Lipschitz or, in some cases, Lipschitz-smooth functions. Some of these methods also come with additional limitations, such as requiring bounded domain assumptions (Khaled et al., 2023) or failing to extend to constrained optimization problems (Defazio & Mishchenko, 2023; Mishchenko & Defazio, 2024).

| Method | Universal | Constraints | Unbounded domain | No search | Stochastic |
|---|---|---|---|---|---|
| DoG (Ivgi et al., 2023) | ✗ | ✓ | ✓ | ✓ | ✓ |
| DoWG (Khaled et al., 2023) | ✗ | ✓ | ✗ | ✓ | ✗ |
| Bisection Search (Carmon & Hinder, 2022) | ✗ | ✓ | ✓ | ✗ | ✓ |
| Prodigy (Mishchenko & Defazio, 2024) | ✗ | ✗ | ✗ | ✓ | ✗ |
| D-Adaptation (Defazio & Mishchenko, 2023) | ✗ | ✗ | ✗ | ✓ | ✗ |
| UGM (Nesterov, 2015) | ✓[(*)] | ✓ | ✓ | ✗ | ✗ |
| DADA (**Ours**) | ✓ | ✓ | ✓ | ✓ | ✗ |

✓[(*)] Note that UGM uses a different definition of universality. They call their method universal because it works for Hölder-smooth functions, which are only a subset of the functions we consider.

Table 1: A comparison of different adaptive algorithms to solve (1). "Universal" means the algorithm achieves problem-dependent convergence rates across convex function classes by exploiting the local growth of the objective near the minimizer. "Constraints" means the algorithm can be applied to constrained problems. "Unbounded domain" means the algorithm can be applied to problems with unbounded feasible sets. "Stochastic" indicates that the algorithm is analyzed in the stochastic setting. "No search" means the algorithm does not use an internal search procedure.

To formalize the discussion, we consider the following optimization problem:

$$f^* := \min_{x \in Q} f(x), \tag{1}$$

where $Q \subseteq \mathbb{R}^d$ is a nonempty closed convex set, and $f \colon \mathbb{R}^d \to \mathbb{R} \cup \{+\infty\}$ is a proper closed convex function that is subdifferentiable on $Q$. We assume that $Q$ is a simple set, meaning that it is possible to efficiently compute the projection onto $Q$. We also assume problem (1) has a solution $x^* \in \operatorname{int} \operatorname{dom} f$. The starting point in our methods is denoted by $x_0$.

**Contributions.** In this paper, we introduce Dual Averaging with Distance Adaptation (DADA), a novel universal gradient method for solving (1). Building on the classical framework of weighted DA (Nesterov, 2005b), DADA incorporates a dynamically adjusted estimate of $D_0 := \|x_0 - x^*\|$, inspired by recent techniques from (Ivgi et al., 2023; Carmon & Hinder, 2022) and further developed in (Khaled et al., 2023), without requiring prior knowledge of problem-specific parameters. Furthermore, our approach applies to both unconstrained problems and those with simple constraints, possibly with unbounded domains. This makes DADA a powerful tool across a wide range of applications.

We start, in Section 2, by presenting our method and outline its foundational structure based on the DA scheme (Nesterov, 2005b). Our main theoretical result, Theorem 1, establishes convergence guarantees for a broad range of function classes.

To demonstrate the versatility and effectiveness of DADA, in Section 3, we provide complexity estimates across several interesting function classes: nonsmooth Lipschitz functions, Lipschitz-smooth functions, Hölder-smooth functions, quasi-self-concordant (QSC) functions, functions with Lipschitz high-order derivative, and $(L_0, L_1)$-smooth functions. These results underscore DADA's ability to deliver competitive performance without knowledge of class-specific parameters.

**Related work.** The development of parameter-free first-order methods has received increasing attention in both optimization and machine learning. A central goal in this line of work is to design algorithms whose performance does not depend on prior knowledge of problem's specific parameters, such as smoothness constants, Lipschitz parameters, or distance to the minimizer—quantities that are rarely known in practice.

Classical approaches to removing stepsize tuning include techniques such as Polyak's stepsize rule (Polyak, 1987) and doubling schedules (Streeter & McMahan, 2012). While effective in certain settings, these strategies either rely on access to the optimal value or introduce additional overhead through repeated restarts. In contrast, more recent parameter-free methods aim to achieve near-optimal performance without requiring such auxiliary procedures.

A large group of recent parameter-free methods is based on AdaGrad-type conditioning (Duchi et al., 2011). These methods adaptively accumulate squared gradient norms to adjust the effective stepsize. This idea underlies several recent distance-adaptation algorithms, including DoG (Ivgi et al., 2023), DoWG (Khaled et al., 2023), D-Adaptation (Defazio & Mishchenko, 2023), and Prodigy (Mishchenko & Defazio, 2024). Although these algorithms achieve parameter-free convergence guarantees for nonsmooth Lipschitz or Lipschitz-smooth objectives, their theoretical rates

do not automatically adapt to broader families of convex functions. We summarize the main properties of the algorithms we compare against in Table 1.

Beyond AdaGrad-type schemes, coin-betting algorithms (Orabona & Pál, 2016) provide adaptive guarantees in online and stochastic optimization by treating learning as a sequential investment game. In a different direction, Carmon and Hinder (Carmon & Hinder, 2022) propose a bisection-based SGD routine that adapts to the unknown smoothness or distance-to-optimum by iteratively solving simpler subproblems. Both coin-betting and bisection approaches are orthogonal to ours but share the goal of eliminating learning rate tuning through adaptation mechanisms.

Another universal method worth noting is Nesterov's Universal Gradient Method (UGM) (Nesterov, 2015), which achieves optimal rates for Hölder-smooth functions via adaptive line search. While UGM is often described as "universal," its scope is limited to smoothness-varying settings and does not extend to broader function classes such as quasi-self-concordant or high-order smooth functions. Moreover, its reliance on internal line search procedures makes it less practical in constrained or composite problems.

**Notation.** In this text, we work in the space $\mathbb{R}^d$ equipped with the standard inner product $\langle \cdot, \cdot \rangle$ and the general Euclidean norm $\|x\| := \langle Bx, x \rangle^{1/2}$, where $B$ is a fixed symmetric positive definite matrix. The corresponding dual norm is defined in the standard way as $\|s\|_* := \max_{\|x\|=1} \langle s, x \rangle = \langle s, B^{-1}s \rangle^{1/2}$. Thus, for any $s, x \in \mathbb{R}^d$, we have the Cauchy-Schwarz inequality $|\langle s, x \rangle| \le \|s\|_* \|x\|$. The Euclidean ball of radius $r > 0$ centered at $x \in \mathbb{R}^d$ is defined as $B(x, r) := \{y \in \mathbb{R}^d : \|y - x\| \le r\}$. For a convex function $f \colon \mathbb{R}^d \to \mathbb{R} \cup \{+\infty\}$, we denote its effective domain as $\operatorname{dom} f := \{x \in \mathbb{R}^d : f(x) < +\infty\}$. The subdifferential of $f$ at a point $x \in \operatorname{dom} f$ is denoted by $\partial f(x)$, and $\nabla f(x) \in \partial f(x)$ denotes a subgradient. We use $\nabla f(x)$ rather than $g \in \partial f(x)$ throughout the paper to keep the notation lightweight.

## 2 DADA METHOD

**Measuring the quality of solution.** Given an approximate solution $x \in Q$ to problem (1) and an arbitrary subgradient $\nabla f(x) \in \partial f(x)$, we measure the suboptimality of $x$ by the distance from $x^*$ to the hyperplane $\{y : \langle \nabla f(x), x - y \rangle = 0\}$:

$$v(x) := \frac{\langle \nabla f(x), x - x^* \rangle}{\|\nabla f(x)\|_*} \quad (\ge 0). \tag{2}$$

This objective is meaningful because minimizing $v(x)$ also reduces the corresponding function residual $f(x) - f^*$. Indeed, there exists the following simple relationship between $v(x)$ and the function residual (Nesterov, 2018, Section 3.2.2) (see also Lemma 3 for the short proof):

$$f(x) - f^* \le \omega(v(x)), \tag{3}$$

where

$$\omega(t) := \max_{x \in B(x^*, t)} f(x) - f^* \tag{4}$$

measures the local growth of $f$ around the solution $x^*$. Note that inequality (3) is nontrivial only when $B(x^*, v(x)) \subseteq \operatorname{dom} f$.

By bounding $\omega(t)$, we can derive convergence-rate estimates that simultaneously apply to a broad range of problem classes (we discuss several examples in Section 3).

**The method.** Our algorithm is based on the general scheme of DA (Nesterov, 2005b) shown in Algorithm 1. Using a standard (sub)gradient method with time-varying coefficients is also possible but requires either short steps by fixing the number of iterations in advance, or paying an extra logarithmic factor in the convergence rate (Nesterov, 2018, Section 3.2.3).

The classical method of Weighted DA (WDA) selects the coefficients $a_k = \frac{\hat{D}_0}{\|g_k\|_*}$ and $\beta_k = \Theta(\sqrt{k})$, where $\hat{D}_0$ is a user-defined estimate of $D_0$. The convergence is guaranteed for any value of $\hat{D}_0$ but one must pay a multiplicative cost of $\rho^2$, where $\rho := \max\{\frac{\hat{D}_0}{D_0}, \frac{D_0}{\hat{D}_0}\}$, if the parameter $D_0$ is

---

**Algorithm 1** General Scheme of DA

---

**Input:** $x_0 \in Q$, number of iterations $T \geq 1$, coefficients $(a_k)_{k=0}^{T-1}$, $(\beta_k)_{k=1}^{T}$ with nondecreasing $\beta_k$
    **for** $k = 0, \ldots, T-1$ **do**
        Compute arbitrary $g_k \in \partial f(x_k)$
        $x_{k+1} = \operatorname{argmin}_{x \in Q} \left\{ \sum_{i=0}^{k} a_i \langle g_i, x - x_i \rangle + \frac{\beta_{k+1}}{2} \|x - x_0\|^2 \right\}$
**Output:** $x_T^* = \operatorname{argmin}_{x \in \{x_0, \ldots, x_{T-1}\}} f(x)$

---

unknown. This cost can be significantly high if $D_0$ is not known almost exactly. To address this issue, we propose DADA, which reduces the cost to a logarithmic term, $\log^2 \rho$, offering a substantial improvement.

Specifically, our approach utilizes the following coefficients:

$$\boxed{a_k = \frac{\bar{r}_k}{\|g_k\|_*}, \quad \beta_k = c\sqrt{k+1}}, \quad \bar{r}_k := \max\{\max_{1 \leq t \leq k} r_t, \bar{r}\}, \quad r_t := \|x_0 - x_t\|, \tag{5}$$

where $\bar{r} > 0$ is a parameter and $c$ is a certain constant to be specified later. In what follows, we assume w.l.o.g. that $g_k \neq 0$ for all $0 \leq k \leq T-1$ since otherwise the exact solution has been found, and the method could be successfully terminated before making $T$ iterations.

As we can see, the main difference between WDA and DADA, is that the latter dynamically adjusts its estimate of $D_0$ by exploiting $r_t$, the distance between $x_t$ and the initial point $x_0$. This idea has been explored in recent works (Carmon & Hinder, 2022; Ivgi et al., 2023), which similarly utilize $r_t$ in various ways. Other methods also attempt to estimate this quantity using alternative strategies, based on DA and the similar principle of employing an increasing sequence of lower bounds for $D_0$ (Defazio & Mishchenko, 2023; Mishchenko & Defazio, 2024).

The convergence guarantees for our method are provided in the result below:

**Theorem 1.** *Consider Algorithm 1 for solving problem* (1) *using the coefficients from* (5) *with* $c > \sqrt{2}$. *Then, for any* $T \geq 1$ *and* $v_T^* := \min_{0 \leq k \leq T-1} v(x_k)$, *it holds that*

$$f(x_T^*) - f^* \leq \omega(v_T^*),$$

*and*

$$v_T^* \leq \frac{eD}{\sqrt{T}} \log \frac{e\bar{D}}{\bar{r}}, \tag{6}$$

*where* $\bar{D} := \max\{\bar{r}, \frac{2c}{c - \sqrt{2}} D_0\}$ *and* $D := \sqrt{2}(cD_0 + \frac{1}{c}\bar{D})$. *Consequently, for a given* $\delta > 0$, *it holds that* $v_T^* \leq \delta$ *whenever* $T \geq T_v(\delta)$, *where*

$$T_v(\delta) := \frac{e^2 D^2}{\delta^2} \log^2 \frac{e\bar{D}}{\bar{r}}.$$

Let us provide a proof sketch for Theorem 1 here and defer the detailed proof to Appendix B. We begin by applying the standard result for DA (Lemma 5), which holds for any choice of coefficients $a_k$ and $\beta_k$:

$$\sum_{i=0}^{k-1} a_i v_i \|g_i\|_* + \frac{\beta_k}{2} D_k^2 \leq \frac{\beta_k}{2} D_0^2 + \sum_{i=0}^{k-1} \frac{a_i^2}{2\beta_i} \|g_i\|_*^2,$$

where $D_i = \|x_i - x^*\|$ and $v_i = v(x_i)$ for all $i \geq 0$. Use the specific choices for $a_k$ and $\beta_k$ as defined in (5), we obtain (see Lemma 6):

$$\sum_{i=0}^{k-1} \bar{r}_i v_i + \frac{c\sqrt{k+1}}{2} D_k^2 \leq \frac{c\sqrt{k+1}}{2} D_0^2 + \frac{\sqrt{k}}{c} \bar{r}_{k-1}^2. \tag{7}$$

Dropping the nonnegative $\bar{r}_i v_i$ from the left-hand side, we can show by induction that $\bar{r}_k$ is uniformly bounded (see Lemma 7):

$$\bar{r}_k \leq \bar{D},$$

where $\bar{D}$ is the constant from Theorem 1. This bound is crucial to our analysis, as we need to eliminate $\bar{r}_{k-1}$ from the right-hand side of (7). Achieving this requires selecting the coefficients precisely as defined in (5), which is the primary difference compared to the standard DA method (Nesterov, 2005b). Next, using the inequality $D_0^2 - D_k^2 \leq 2r_k D_0$, we get

$$\sum_{i=0}^{k-1} \bar{r}_i v_i \leq c\sqrt{k+1}\, r_k D_0 + \frac{\sqrt{k}}{c}\bar{r}_{k-1}^2 \leq \left(cD_0 + \frac{1}{c}\bar{D}\right)\bar{r}_k\sqrt{k+1}.$$

After establishing this, the rest of the proof follows straightforwardly by dividing both sides by $\sum_{i=0}^{k-1} \bar{r}_i$ and applying the following inequality (valid for any nondecreasing sequence $\bar{r}_k$, see Lemma 2):

$$\min_{1 \leq k \leq T} \frac{\bar{r}_k}{\sum_{i=0}^{k-1} \bar{r}_i} \leq \frac{(\frac{\bar{r}_T}{\bar{r}_0})^{\frac{1}{T}} \log \frac{e\bar{r}_T}{\bar{r}_0}}{T}.$$

This gives us

$$v_T^* \leq \frac{D}{\sqrt{T}} \left(\frac{\bar{D}}{\bar{r}}\right)^{\frac{1}{T}} \log \frac{e\bar{D}}{\bar{r}},$$

which is almost (6) except for the extra factor of $(\frac{\bar{D}}{\bar{r}})^{\frac{1}{T}}$. This extra factor, however, is rather weak as it can be upper bounded by a constant (say, $e \equiv \exp(1)$) whenever $T \geq \log \frac{\bar{D}}{\bar{r}}$. The case of $T \leq \log \frac{\bar{D}}{\bar{r}}$ is not interesting since then (6) holds trivially because, for any $k \geq 0$, in view of (2) and Lemma 7, we have $v_k \leq D_k \leq D$. According to Theorem 1, our method converges for any $c > \sqrt{2}$. To obtain the smallest complexity bound (up to logarithmic factors), the value that minimizes this bound is $c = 2\sqrt{2}$. A more detailed discussion of this choice is provided in Appendix C.

## 3 UNIVERSALITY OF DADA: EXAMPLES OF APPLICATIONS

Let us demonstrate that our method is *universal* in the sense that it simultaneously works for multiple problem classes without the need for choosing different parameters for each of these function classes. For simplicity, we assume that $\nabla f(x^*) = 0$ (this happens, in particular, when our problem (1) is unconstrained) and measure the $\epsilon$-accuracy in terms of the function residual. This assumption is made only to keep the discussion clean and readable, and it also reflects an important practical setting (unconstrained problems, or constrained problems with $x^*$ in the interior of $Q$). The general constrained case, where $\nabla f(x^*) \neq 0$, is covered in Appendix D; there, for some function classes (e.g., Lipschitz-smooth), the complexity bounds include an additional $\|\nabla f(x^*)\|_*^2/\epsilon^2$ term that may affect the rate. We also assume, for simplicity, that the objective function satisfies all necessary inequalities on the entire space, but all our results still hold if they are satisfied only locally at $x^*$ (see Appendix D). To simplify the notation, we also denote $\log_+ t := 1 + \log t$ and $\bar{D}_0 := \max\{\bar{r}, \|x_0 - x^*\|\}$, where $\bar{r}$ is the parameter of our method.

**Nonsmooth Lipschitz functions.** This function class is defined by the inequality

$$|f(x) - f(y)| \leq L_0\|x - y\|$$

for all $x, y \in \mathbb{R}^d$. For this problem class, DADA requires at most (see Corollary 10)

$$O\left(\frac{L_0^2 \bar{D}_0^2}{\epsilon^2} \log_+^2 \frac{\bar{D}_0}{\bar{r}}\right)$$

oracle calls to reach $\epsilon$-accuracy, which matches the standard complexity of (sub)gradient methods (Nesterov, 2005b; 2018), up to an extra logarithmic factor. Note that a polylogarithmic factor in $\frac{\bar{D}_0}{\bar{r}}$ appears in the complexity bounds of all distance-adaptation methods (Defazio & Mishchenko, 2023; Ivgi et al., 2023; Khaled et al., 2023; Mishchenko & Defazio, 2024).

**Lipschitz-smooth functions.** Another important class of functions are those with Lipschitz gradient:

$$\|\nabla f(x) - \nabla f(y)\|_* \le L_1 \|x - y\|$$

for all $x, y \in \mathbb{R}^d$. In this case, the complexity of our method is (see Corollary 13)

$$O\left(\frac{L_1 \bar{D}_0^2}{\epsilon} \log_+^2 \frac{\bar{D}_0}{\bar{r}}\right).$$

This coincides with the standard complexity of the (nonaccelerated) gradient method on Lipschitz-smooth functions (Nesterov, 2018, Section 3) up to an extra logarithmic factor.

Note that the complexity of DADA is slightly worse than that of the classical gradient method with line search (Nesterov, 2015), which achieves a complexity bound of $O\left(\frac{L_1 D_0^2}{\epsilon} + \log\left|\frac{L_1}{\hat{L}_1}\right|\right)$, where $\hat{L}_1$ is the initial guess for $L_1$. The difference is that the logarithmic factor in the latter estimate appears in an additive way instead of multiplicative.

**Hölder-smooth functions.** The previous two examples are subclasses of the more general class of Hölder-smooth functions. It is defined by the following inequality:

$$\|\nabla f(x) - \nabla f(y)\|_* \le H_\nu \|x - y\|^\nu$$

for all $x, y \in \mathbb{R}^d$, where $\nu \in [0, 1]$ and $H_\nu \ge 0$. Therefore, for $\nu = 0$, we get functions with bounded variation of subgradients (which contains all Lipschitz functions) and for $\nu = 1$ we get Lipschitz-smooth functions.

The complexity of DADA on this problem class is (see Corollary 16)

$$O\left(\left[\frac{H_\nu}{\epsilon}\right]^{\frac{2}{1+\nu}} \bar{D}_0^2 \log_+^2 \frac{\bar{D}_0}{\bar{r}}\right).$$

This is similar to the $O\left(\left[\frac{H_\nu}{\epsilon}\right]^{\frac{2}{1+\nu}} D_0^2 + \log\left|\frac{H_\nu^{\frac{2}{1+\nu}}}{\hat{L}\epsilon^{\frac{1-\nu}{1+\nu}}}\right|\right)$ complexity of the universal (nonaccelerated) gradient method with line search (GM-LS) (Nesterov, 2015), where $\hat{L}$ is the parameter of the method. Again, the complexity of GM-LS is slightly better since the logarithmic factor is additive (and not multiplicative). However, GM-LS is not guaranteed to work (well) on other problem classes such as those we consider next.

**Functions with Lipschitz high-order derivative.** This class is a generalization of the Lipschitz-smooth class. Functions in this class are $p$ times differentiable, and have the property that their $p$th derivative ($p \ge 2$) is Lipschitz, i.e., for all $x, y \in \mathbb{R}^d$, we have

$$\|\nabla^p f(x) - \nabla^p f(y)\| \le L_p \|x - y\|,$$

where the $\|\cdot\|$ norm in the left-hand side is the usual operator norm of a symmetric $p$-linear operator: $\|A\| = \max_{h \in \mathbb{R}^d: \|h\|=1} \|A[h]^p\|$. For example, the $p$th power of the Euclidean norm is an example of a function in this class (see (Rodomanov & Nesterov, 2019)). The complexity of DADA on this problem class is (see Corollary 19)

$$O\left(\left[\max_{2 \le i \le p}\left[\frac{p}{i!} \frac{\|\nabla^i f(x^*)\|_*}{\epsilon}\right]^{\frac{2}{i}} + \left[\frac{L_p}{p! \, \epsilon}\right]^{\frac{2}{p+1}}\right] \bar{D}_0^2 \log_+^2 \frac{\bar{D}_0}{\bar{r}}\right).$$

Although line-search gradient methods might be better for Hölder-smooth problems, to our knowledge, they are not known to attain comparable bounds on this function class.

**Quasi-self-concordant (QSC) functions (Bach, 2010).** A function $f$ is called QSC with parameter $M \ge 0$ if it is three times continuously differentiable and the following inequality holds for any $x, u, v \in \mathbb{R}^d$:

$$\nabla^3 f(x)[u, u, v] \le M \langle \nabla^2 f(x)u, u \rangle \|v\|. \tag{8}$$

For example, the exponential, logistic, and softmax functions are QSC; for more details and other examples, see (Doikov, 2023). When applied to a QSC function, our method has the following complexity (Corollary 23):

$$O\left(\left[M^2\bar{D}_0^2 + \frac{\|\nabla^2 f(x^*)\|\bar{D}_0^2}{\epsilon}\right]\log_+^2\frac{\bar{D}_0}{\bar{r}}\right).$$

In terms of comparisons, second-order methods, such as those explored in (Doikov, 2023), are more powerful for minimizing QSC functions, as they leverage additional curvature information. Their complexity bound, in terms of queries to the second-order oracle, is $O(M\hat{D}_0\log\frac{F_0}{\epsilon} + \log\frac{\hat{D}_0 g_0}{\epsilon F_0})$, where $F_0 = f(x_0) - f^*$, $\hat{D}_0$ is the diameter of the initial sublevel set, and $g_0 = \|\nabla f(x_0)\|_*$ (see (Doikov, 2023, Corollary 3.4)). However, each iteration of these methods is significantly more expensive.

To our knowledge, the QSC class has not been previously studied in the context of first-order methods. The only other first-order methods for which one can prove similar bounds are the nonadaptive variants of our scheme, namely the normalized gradient method (NGM) from (Nesterov, 2018, Section 5) and the recent improvement of this algorithm for constrained problems (Nesterov, 2024).

$(L_0, L_1)$-**smooth functions.** As introduced in (Zhang et al., 2020), a function $f$ is said to be $(L_0, L_1)$-smooth if for all $x \in \mathbb{R}^d$, we have

$$\|\nabla^2 f(x)\| \le L_0 + L_1\|\nabla f(x)\|_*.$$

The complexity of DADA on this class is (see Corollary 26)

$$O\left(\left[L_1^2\bar{D}_0^2 + \frac{L_0\bar{D}_0^2}{\epsilon}\right]\log_+^2\frac{\bar{D}_0}{\bar{r}}\right).$$

Up to the extra logarithmic factor, this matches the complexity of NGM from (Vankov et al., 2024), with the distinction that their approach is less robust to the initial guess of $D_0$. Specifically, the penalty for underestimating it in the latter method is a multiplicative factor of $\rho^2 := \frac{D_0^2}{\bar{r}^2}$ while in our method this factor is logarithmic: $\log_+^2\rho$.

## 4 EXPERIMENTS

To evaluate the efficiency of our proposed method, DADA, we conduct a series of experiments on convex optimization problems. Our goal is to demonstrate the effectiveness of DADA in achieving competitive performance across various function classes *without any hyperparameter tuning*.

We compare DADA against state-of-the-art distance-adaptation algorithms, namely, DoG (Ivgi et al., 2023) and Prodigy (Mishchenko & Defazio, 2024), using their official implementations without any modifications. We also consider the Universal Gradient Method (UGM) from (Nesterov, 2015) and the classical Weighted Dual Averaging (WDA) method (Nesterov, 2005b). For UGM, we choose the initial value of the line-search parameter $L_0 = 1$ and set the target accuracy to $\epsilon = 10^{-6}$. For WDA, we use the coefficients $a_k = \frac{D_0}{\|g_k\|_*}$ and $\beta_k = \sqrt{k}$, where $D_0 = \|x_0 - x^*\|$.

For each method, we plot the best function value among all the test points generated by the algorithm against the number of first-order oracle calls. We set the starting point to $x_0 = (1, \ldots, 1)$ and select the initial guess for the distance to the solution as $\bar{r} = \delta(1 + \|x_0\|)$. This choice ensures a fair comparison between DADA and DoG (Ivgi et al., 2023), as DoG employs a similar initialization strategy. In all experiments, we fix $\delta = 10^{-6}$. Additionally, we conduct a separate experiment to evaluate the sensitivity of DADA to the choice of $\delta$.

We have several experiments on different problem classes. However, due to space constraints, we present only a single representative experiment in this section. The remaining experiments can be found in Appendix E.

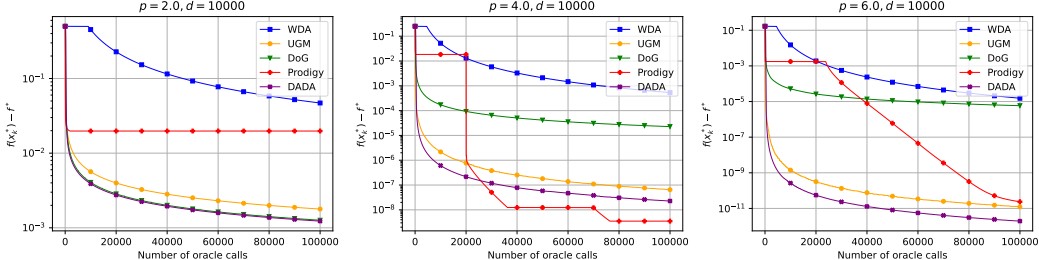

Figure 1: Comparison of different methods on the worst-case function.

**Worst-case function.** As an example of a function with Lipschitz high-order derivative, we consider the following worst-case problem from (Doikov et al., 2024):

$$\min_{x \in \mathbb{R}^d} \left\{ f(x) \coloneqq \frac{1}{p} \sum_{i=1}^{d-1} |x^{(i)} - x^{(i+1)}|^p + \frac{1}{p} |x^{(d)}|^p \right\}, \tag{9}$$

where $p \geq 2$, and $x^{(i)}$ is the $i$th element of $x$. The optimal point in this problem is $x^* = 0$.

As illustrated in Fig. 1, nearly all methods exhibit similar performance when $p = 2$, except for Prodigy whose convergence becomes slow after a few initial iterations. While Prodigy eventually reaches a similar accuracy to the best methods, it is much slower at the beginning of the process. As $p$ increases, our method converges significantly faster than DoG. We suspect that this improvement arises because our method adapts to the high-order smoothness of the function, whereas DoG's convergence rate remains unchanged and does not take advantage of this property.

In contrast, both DADA and UGM demonstrate stable and consistent performance across different values of $p$, with DADA performing slightly better than UGM.

## 5 DISCUSSION

**Comparison with recent distance-adaptation methods.** Let us briefly compare our method with several recently proposed parameter-free algorithms, namely, DoG (Ivgi et al., 2023), DoWG (Khaled et al., 2023), D-Adaptation (Defazio & Mishchenko, 2023) and Prodigy (Mishchenko & Defazio, 2024).

To begin, we clarify the key differences between our method and other existing gradient methods using the distance-adaptation technique. One immediate difference is that we use DA instead of the classical (sub)gradient method employed by DoG. We could also instantiate our approach using the standard subgradient method instead of DA. However, doing so would either require fixing the number of iterations in advance or would worsen the overall complexity by an additional poly-logarithmic factor in the target accuracy. However, the most significant difference lies in how the sequence of gradients is handled. In contrast to existing distance-adaptation methods, which follow the AdaGrad (Duchi et al., 2011) principle of accumulating squared gradient norms, our method simply normalizes $g_k$ by its own norm. This modification makes our method universal, ensuring that $v_T^*$—the distance from $x^*$ to the supporting hyperplane—converges to zero, which is not known to be the case for DoG, even for deterministic problems.

Both DoG and DoWG employ a similar approach to estimate $D_0 = \|x_0 - x^*\|$ and achieve comparable convergence rates for Lipschitz-smooth and nonsmooth functions. Similarly to our approach, DoWG considers only the deterministic case, but with an additional assumption of the bounded feasible set. They have a different definition of universality, considering only Lipschitz-smooth and nonsmooth settings.

Furthermore, to the best of our knowledge, these D-Adaptation and Prodigy have not been extended to constrained optimization. Nonetheless, their methods yield notable results in experiments, demonstrating strong empirical performance.

It is important to emphasize that the advantage of our method over DoG does not lie in guaranteeing convergence. Indeed, (Ivgi et al., 2023, Theorem 1) shows that DoG asymptotically converges to a minimizer for any convex function, with a complexity of $\widetilde{O}\left(\frac{L_R^2 \bar{D}_0^2}{\epsilon^2}\right)$, where $L_R = \max_{x \in B(x_0, R)} \|\nabla f(x)\|_*$ for $R = 3\bar{D}_0$, and $\epsilon$ denotes the target accuracy in the function value. However, this complexity bound has a critical drawback—it remains inversely proportional to $\epsilon^2$ across *all* function classes, which is not the case for our method. For illustration, consider the setting where $f$ has an $L_2$-Lipschitz Hessian. Further, assume for simplicity that the problem is unconstrained and that $\|\nabla^2 f(x^*)\|$ is zero (or negligibly small). In this case, the above complexity bound for DoG becomes[1] $\widetilde{O}\left(\frac{L_2^2 \bar{D}_0^6}{\epsilon^2}\right)$, which is substantially worse than $\widetilde{O}\left(\frac{L_2^{2/3} \bar{D}_0^2}{\epsilon^{2/3}}\right)$ for DADA (see Corollary 19). Thus, in comparison to DoG, our method provides significantly stronger efficiency guarantees and exhibits automatic acceleration under favorable conditions for a considerably broader family of function classes.

**Conclusion.** We proposed DADA, a new adaptive and universal optimization method that extends the classical Dual Averaging algorithm with a novel distance adaptation mechanism. Our method achieves competitive rates across a wide class of convex problems—including Lipschitz, Lipschitz-smooth, Hölder-smooth, quasi-self-concordant (QSC), and $(L_0, L_1)$-smooth functions—without requiring parameter tuning or knowledge of smoothness constants. In contrast to recent approaches such as DoG, DoWG, D-Adaptation, and Prodigy, DADA seamlessly accommodates both constrained and unconstrained settings, and does so without requiring restarts or line searches.

DADA provides a unified and adaptive framework for convex optimization with convergence guarantees under minimal assumptions. Future work includes extending DADA to stochastic and non-convex optimization, and evaluating its empirical performance in large-scale learning tasks.

ACKNOWLEDGMENTS

This work is supported in part by the Institute for Computing, Information and Cognitive Systems (ICICS) at UBC.

---

[1]Here, we use the fact that for functions with Lipschitz Hessian, $L_R = O\left(L_1^* R + \frac{L_2}{2} R^2\right)$, where $L_1^* = \|\nabla^2 f(x^*)\|$.

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

# A  AUXILIARY RESULTS

The following result has been established in prior works such as (Liu & Zhou, 2023, Lemma 30). We include the proof here for the reader's convenience.

**Lemma 2.** *Let $(d_i)_{i=0}^{\infty}$ be a positive nondecreasing sequence. Then for any $T \geq 1$,*

$$\min_{1 \leq k \leq T} \frac{d_k}{\sum_{i=0}^{k-1} d_i} \leq \frac{(\frac{d_T}{d_0})^{\frac{1}{T}} \log \frac{e d_T}{d_0}}{T}.$$

*Proof.* Let $A_k := \frac{1}{d_k} \sum_{i=0}^{k-1} d_i$ for each $k \geq 0$ (so that $A_0 = 0$). Then, for each $k \geq 0$, we have

$$d_{k+1} A_{k+1} - d_k A_k = d_k,$$

which implies that

$$\frac{d_k}{d_{k+1}} = A_{k+1} - \frac{d_k}{d_{k+1}} A_k = A_{k+1} - A_k + \left(1 - \frac{d_k}{d_{k+1}}\right) A_k.$$

Summing up these identities for all $0 \leq k \leq T - 1$, we get

$$S_T := \sum_{k=0}^{T-1} \frac{d_k}{d_{k+1}} = A_T + \sum_{k=0}^{T-1} \left(1 - \frac{d_k}{d_{k+1}}\right) A_k \leq A_T^*(1 + T - S_T),$$

where $A_T^* = \max_{0 \leq k \leq T} A_k \equiv \max_{1 \leq k \leq T} A_k$ and we have used the fact that $(d_i)_{i=0}^{\infty}$ is nondecreasing. Hence,

$$A_T^* \geq \frac{S_T}{1 + T - S_T}.$$

Applying now the AM-GM inequality and denoting $\gamma_T = (\frac{d_0}{d_T})^{\frac{1}{T}} (\in (0, 1])$, we can further estimate $S_T \geq T\gamma_T$, giving us

$$A_T^* \geq \frac{T\gamma_T}{1 + T(1 - \gamma_T)}.$$

Thus,

$$\min_{1 \leq k \leq T} \frac{d_k}{\sum_{i=0}^{k-1} d_i} = \frac{1}{A_T^*} \leq \frac{\frac{1}{\gamma_T}(1 + T(1 - \gamma_T))}{T}.$$

Estimating further $T(1 - \gamma_T) \leq -T \log \gamma_T \equiv \log \frac{1}{\gamma_T^T}$ and substituting the definition of $\gamma_T$, we get the claim. $\qquad \square$

The following lemma is a classical result from (Nesterov, 2018, Lemma 3.2.1).

**Lemma 3.** *Let $x \in \mathrm{dom}\, f$ be such that $f$ is subdifferentiable on $x$. Then, we have $f(x) - f^* \leq \omega(v(x))$, where $\omega(\cdot)$ and $v(\cdot)$ are defined as in (2) and (4) (with $\nabla f(x)$ being an arbitrary subgradient from $\partial f(x)$).*

*Proof.* Let $\bar{x}$ denote the orthogonal projection of $x^*$ onto the supporting hyperplane $\{y : \langle \nabla f(x), x - y \rangle = 0\}$:

$$\bar{x} = x^* + v(x) \frac{B^{-1} \nabla f(x)}{\|\nabla f(x)\|_*}.$$

Then, $\langle \nabla f(x), \bar{x} - x \rangle = 0$, and $\|\bar{x} - x^*\| = v(x)$. Therefore,

$$f(x) \leq f(\bar{x}) + \langle \nabla f(x), \bar{x} - x \rangle = f(\bar{x}),$$

and hence,

$$f(x) - f^* \leq f(\bar{x}) - f^* \leq \omega(\|\bar{x} - x^*\|) = \omega(v(x)). \qquad \square$$

**Lemma 4.** *Consider the nonnegative sequence $(d_k)_{k=0}^{\infty}$ that satisfies, for each $k \geq 0$,*

$$d_{k+1} \leq \max\{d_k, R + \gamma d_k\},$$

*where $0 \leq \gamma < 1$ and $R \geq 0$ are certain constants. Then, for any $k \geq 0$, we have*

$$d_k \leq \max\left\{\frac{1}{1-\gamma}R, d_0\right\}.$$

*Proof.* We use induction to prove that $d_k \leq D$ for a certain constant $D$ to be determined later. To ensure that this relation holds for $k = 0$, we need to choose $D \geq d_0$. Let us now suppose that our relation has already been proved for some $k \geq 0$ and let us prove it for the next index $k + 1$. Using the induction hypothesis and the given inequality, we obtain

$$d_{k+1} \leq \max\{d_k, R + \gamma d_k\} \leq \max\{D, R + \gamma D\}.$$

To prove that the right-hand side is $\leq D$, we need to ensure that $R + \gamma D \leq D$, which means that we need to choose $D \geq \frac{1}{1-\gamma}R$. Combining this requirement with that from the base of induction, we see that we can choose $D = \max\{\frac{1}{1-\gamma}R, d_0\}$. □

## B  PROOF OF THEOREM 1

**Lemma 5.** *In Algorithm 1, for any $1 \leq k \leq T$, it holds that*

$$\sum_{i=0}^{k-1} a_i \langle g_i, x_i - x^* \rangle + \frac{\beta_k}{2}\|x_k - x^*\|^2 \leq \frac{\beta_k}{2}\|x_0 - x^*\|^2 + \sum_{i=0}^{k-1} \frac{a_i^2}{2\beta_i}\|g_i\|_*^2,$$

*where $\beta_0$ is an arbitrary coefficient in $(0, \beta_1]$.*

*Proof.* For any $0 \leq k \leq T$, define the function $\psi_k(x)$ as follows:

$$\psi_k(x) := \sum_{i=0}^{k-1} a_i \langle g_i, x - x_i \rangle + \frac{\beta_k}{2}\|x - x_0\|^2,$$

so that $\psi_0(x) = \frac{\beta_0}{2}\|x - x_0\|^2$ (with $\beta_0$ as defined in the statement). Note that $\psi_k$ is a $\beta_k$-strongly convex function and $x_k$ is its minimizer. Hence, for any $x \in Q$ and $0 \leq k \leq T$, we have

$$\psi_k(x) \geq \psi_k^* + \frac{\beta_k}{2}\|x - x_k\|^2, \tag{10}$$

where $\psi_k^* := \psi_k(x_k)$. Consequently,

$$
\begin{aligned}
\psi_{k+1}^* = \psi_{k+1}(x_{k+1}) &= \psi_k(x_{k+1}) + a_k \langle g_k, x_{k+1} - x_k \rangle + \frac{\beta_{k+1} - \beta_k}{2}\|x_{k+1} - x_0\|^2 \\
&\geq \psi_k^* + \frac{\beta_k}{2}\|x_{k+1} - x_k\|^2 + a_k \langle g_k, x_{k+1} - x_k \rangle + \frac{\beta_{k+1} - \beta_k}{2}\|x_{k+1} - x_0\|^2 \\
&\geq \psi_k^* + \frac{\beta_k}{2}\|x_{k+1} - x_k\|^2 + a_k \langle g_k, x_{k+1} - x_k \rangle \geq \psi_k^* - \frac{a_k^2}{2\beta_k}\|g_k\|_*^2.
\end{aligned}
$$

Telescoping these inequalities and using the fact that $\psi_0^* = 0$, we obtain

$$\psi_k^* \geq -\sum_{i=0}^{k-1} \frac{a_i^2}{2\beta_i}\|g_i\|_*^2.$$

Combining this inequality with the definition of $\psi_k$ and (10), we thus obtain

$$\sum_{i=0}^{k-1} a_i \langle g_i, x^* - x_i \rangle + \frac{\beta_k}{2}\|x_0 - x^*\|^2 = \psi_k(x^*) \geq \psi_k^* + \frac{\beta_k}{2}\|x_k - x^*\|^2$$

$$\geq -\sum_{i=0}^{k-1} \frac{a_i^2}{2\beta_i}\|g_i\|_*^2 + \frac{\beta_k}{2}\|x_k - x^*\|^2.$$

Rearranging, we get the claim. □

**Lemma 6.** *Consider Algorithm 1 using the coefficients defined in* (5). *Then, the following inequality holds for all* $1 \leq k \leq T$:

$$\sum_{i=0}^{k-1} \bar{r}_i v_i + \frac{c\sqrt{k+1}}{2} D_k^2 \leq \frac{c\sqrt{k+1}}{2} D_0^2 + \frac{\sqrt{k}}{c} \bar{r}_{k-1}^2,$$

*where* $D_k = \|x_k - x^*\|$ *and* $v_i := v(x_i)$.

*Proof.* Applying Lemma 5 and the definition of $v_i$, we obtain

$$\sum_{i=0}^{k-1} a_i v_i \|g_i\|_* + \frac{\beta_k}{2} D_k^2 \leq \frac{\beta_k}{2} D_0^2 + \sum_{i=0}^{k-1} \frac{a_i^2}{2\beta_i} \|g_i\|_*^2.$$

Substituting our choice of the coefficients given by (5), we get

$$\sum_{i=0}^{k-1} \bar{r}_i v_i + \frac{c\sqrt{k+1}}{2} D_k^2 \leq \frac{c\sqrt{k+1}}{2} D_0^2 + \frac{1}{2c} \sum_{i=0}^{k-1} \frac{\bar{r}_i^2}{\sqrt{i+1}} \leq \frac{c\sqrt{k+1}}{2} D_0^2 + \frac{\sqrt{k}}{c} \bar{r}_{k-1}^2,$$

where we have used the fact that $\bar{r}_k$ is nondecreasing and $\sum_{i=0}^{k-1} \frac{1}{\sqrt{i+1}} \leq 2\sqrt{k}$. $\square$

**Lemma 7.** *Consider Algorithm 1 using the coefficients defined in* (5) *and assume that* $c > \sqrt{2}$. *Then, we have the following inequalities for all* $0 \leq k \leq T$:

$$\bar{r}_k \leq \bar{D}, \qquad D_k \leq D_0 + \frac{\sqrt{2}}{c} \bar{D},$$

*where* $\bar{D} := \max\{\bar{r}, \frac{2c}{c-\sqrt{2}} D_0\}$ *and* $D_k := \|x_k - x^*\|$.

*Proof.* Both bounds are clearly valid for $k = 0$, so it suffices to consider only the case when $1 \leq k \leq T$.

Applying Lemma 6, dropping the nonnegative $\bar{r}_i v_i$ from the left-hand side and rearranging, we obtain

$$D_k^2 \leq D_0^2 + \frac{2\sqrt{k}}{c^2\sqrt{k+1}} \bar{r}_{k-1}^2 \leq D_0^2 + \frac{2}{c^2} \bar{r}_{k-1}^2.$$

Consequently,

$$D_k \leq D_0 + \frac{\sqrt{2}}{c} \bar{r}_{k-1}. \tag{11}$$

Therefore,

$$r_k \equiv \|x_k - x_0\| \leq D_k + D_0 \leq 2D_0 + \frac{\sqrt{2}}{c} \bar{r}_{k-1}.$$

Hence,

$$\bar{r}_k \equiv \max\{\bar{r}_{k-1}, r_k\} \leq \max\left\{\bar{r}_{k-1}, 2D_0 + \frac{\sqrt{2}}{c} \bar{r}_{k-1}\right\}.$$

Since $k \geq 1$ was allowed to be arbitrary, we can apply Lemma 4 to conclude that

$$\bar{r}_k \leq \max\left\{\bar{r}, \frac{2}{1 - \frac{\sqrt{2}}{c}} D_0\right\} = \max\left\{\bar{r}, \frac{2c}{c - \sqrt{2}} D_0\right\} \equiv \bar{D}.$$

This proves the first part of the claim.

Substituting the already proved bound on $\bar{r}_k$ into (11), we obtain the claimed upper bound on $D_k$. $\square$

We are now ready to prove the main result.

*Proof of Theorem 1.* Let $T \geq 1$ be arbitrary. According to Lemma 3 and the fact that $\omega(\cdot)$ is nondecreasing, we can write

$$f(x_T^*) - f^* = \min_{0 \leq k \leq T-1}[f(x_k) - f^*] \leq \min_{0 \leq k \leq T-1} \omega(v_k) = \omega(v_T^*),$$

where $v_k := v(x_k)$ and $v_T^* := \min_{0 \leq k \leq T-1} v_k$. This proves the first part of the claim.

Let us now estimate the rate of convergence of $v_T^*$. To that end, let us fix an arbitrary $1 \leq k \leq T$. In view of Lemma 6, we have

$$\sum_{i=0}^{k-1} \bar{r}_i v_i \leq \frac{c\sqrt{k+1}}{2}(D_0^2 - D_k^2) + \frac{\sqrt{k}}{c}\bar{r}_{k-1}^2,$$

where $D_k = \|x_k - x^*\|$. Note that

$$D_0^2 - D_k^2 \equiv \|x_0 - x^*\|^2 - \|x_k - x^*\|^2 = (\|x_0 - x^*\| - \|x_k - x^*\|)(\|x_0 - x^*\| + \|x_k - x^*\|)$$
$$\leq 2\|x_k - x_0\|\|x_0 - x^*\| \equiv 2r_k D_0.$$

Therefore, we can continue as follows:

$$\sum_{i=0}^{k-1} \bar{r}_i v_i \leq c\sqrt{k+1} r_k D_0 + \frac{\sqrt{k}}{c}\bar{r}_{k-1}^2 \leq \left(cD_0 + \frac{1}{c}\bar{r}_{k-1}\right)\sqrt{k+1}\,\bar{r}_k$$

$$\leq \left(cD_0 + \frac{1}{c}\bar{D}\right)\sqrt{k+1}\,\bar{r}_k = D\sqrt{\frac{k+1}{2}}\,\bar{r}_k,$$

where the second inequality is due to the fact that $\bar{r}_k = \max\{\bar{r}_{k-1}, r_k\}$, the final inequality is due to Lemma 7, and the constants $\bar{D}$ and $D$ are as defined in the statement. Hence,

$$v_k^* \equiv \min_{0 \leq i \leq k-1} v_i \leq \frac{\sum_{i=0}^{k-1} \bar{r}_i v_i}{\sum_{i=0}^{k-1} \bar{r}_i} \leq \frac{\bar{r}_k}{\sum_{i=0}^{k-1} \bar{r}_i} D\sqrt{\frac{k+1}{2}}.$$

Letting now $k^* = \operatorname{argmin}_{1 \leq k \leq T} \frac{\bar{r}_k}{\sum_{i=0}^{k-1} \bar{r}_i}$ and using Lemma 2, we obtain

$$v_T^* \leq v_{k^*}^* \leq \frac{D\sqrt{\frac{k^*+1}{2}}}{T}\left(\frac{\bar{r}_T}{\bar{r}}\right)^{\frac{1}{T}}\log\frac{e\bar{r}_T}{\bar{r}} \leq \frac{D}{\sqrt{T}}\left(\frac{\bar{D}}{\bar{r}}\right)^{\frac{1}{T}}\log\frac{e\bar{D}}{\bar{r}},$$

where we have used the fact that $k^* + 1 \leq T + 1 \leq 2T$ (since $1 \leq k^* \leq T$) and that $\bar{r}_T \leq \bar{D}$ (see Lemma 7). This proves (6) in the case when $T \geq \log\frac{\bar{D}}{\bar{r}}$ since then we can further bound $(\frac{\bar{D}}{\bar{r}})^{\frac{1}{T}} \equiv \exp(\frac{1}{T}\log\frac{\bar{D}}{\bar{r}}) \leq e$.

On the other hand, by the definition of $v_k$ and Lemma 7, we always have the following trivial inequality for any $0 \leq k \leq T - 1$:

$$v_k \equiv \frac{\langle \nabla f(x_k), x_k - x^* \rangle}{\|\nabla f(x_k)\|_*} \leq D_k \leq D_0 + \frac{\sqrt{2}}{c}\bar{D} \leq D.$$

This means that (6) is also satisfied in the case when $T \leq \log\frac{\bar{D}}{\bar{r}}$ since then $\frac{eD}{\sqrt{T}}\log\frac{e\bar{D}}{\bar{r}} \geq \frac{D}{\sqrt{T}}\log\frac{\bar{D}}{\bar{r}} \geq D\sqrt{T} \geq D$ (we still consider $T \geq 1$). The proof of (6) is now finished.

The final part of the claim readily follows from (6). □

## C  HOW TO CHOOSE THE CONSTANT $c$

According to Theorem 1, our method converges for any $c > \sqrt{2}$. However, the choice of $c$ can influence the constant factor in the complexity of DADA. Hence, our goal here is to find the optimal constant $c$ that minimizes $T_v(\delta)$. To determine this $c$, let $\bar{r}$ be sufficiently small, so that

$$\bar{D} \equiv \max\left\{\bar{r}, \frac{2c}{c - \sqrt{2}}D_0\right\} = \frac{2c}{c - \sqrt{2}}D_0.$$

Then, disregarding the logarithmic factors, due to their minimal impact on the complexity of our method, we can determine the optimal constant $c$ that minimizes

$$D \equiv \sqrt{2}\left(cD_0 + \frac{1}{c}\bar{D}\right) = \sqrt{2}\left(c + \frac{2}{c - \sqrt{2}}\right)D_0.$$

This is the value

$$c = 2\sqrt{2}. \tag{12}$$

For this optimal choice of $c$, we get $\bar{D} = \max\{\bar{r}, 4D_0\}$ and $D = 4D_0 + \frac{1}{2}\bar{D}$, so the complexity of our method given by Theorem 1 is

$$T_v(\delta) = \frac{e^2(4D_0 + \frac{1}{2}\bar{D})^2}{\delta^2}\log^2\frac{e\bar{D}}{\bar{r}}.$$

# D  CONVERGENCE OF DADA ON VARIOUS PROBLEM CLASSES

In this section, we analyze the complexity of DADA across different problem classes. To achieve this, we first establish bounds on the growth function:

$$\omega(t) = \max_{x \in B(x^*, t)} f(x) - f^*,$$

and determine the threshold $t$ such that $\omega(t) \le \epsilon$ for a given $\epsilon$. Subsequently, we combine these results with the complexity bound $T(\delta)$ derived in Theorem 1, enabling us to estimate the oracle complexity of DADA for finding an $\epsilon$-solution in terms of the function residual.

## D.1  NONSMOOTH LIPSCHITZ FUNCTIONS

**Assumption 8.** *The function $f$ in problem* (1) *is locally Lipschitz at $x^*$. Specifically, for any $x \in B(x^*, \rho)$, the following inequality holds:*

$$f(x) - f^* \le L_0\|x - x^*\|, \tag{13}$$

*where $L_0, \rho > 0$ are fixed constants.*

**Lemma 9.** *Let $f$ be locally $L_0$-Lipschitz at $x^*$ (Assumption 8). Then, $\omega(t) \le \epsilon$ for any given $\epsilon > 0$ whenever $t \le \delta(\epsilon)$, where*

$$\delta(\epsilon) := \min\left\{\frac{\epsilon}{L_0}, \rho\right\}.$$

*Proof.* According to (13), for any $0 \le t \le \rho$, we have

$$\omega(t) \le L_0 t.$$

Making the right-hand side $\le \epsilon$, we get the claim. □

Combining Theorem 1 and Lemma 9, we get the following complexity result.

**Corollary 10.** *Consider problem* (1) *under Assumption 8. Let Algorithm 1 with coefficients* (5) *be applied for solving this problem. Then, $f(x_T^*) - f^* \le \epsilon$ for any given $\epsilon > 0$ whenever $T \ge T(\epsilon)$, where*

$$T(\epsilon) = \max\left\{\frac{L_0^2}{\epsilon^2}, \frac{1}{\rho^2}\right\}e^2D^2\log^2\frac{e\bar{D}}{\bar{r}},$$

*and the constants $D$ and $\bar{D}$ are as defined in Theorem 1.*

## D.2 LIPSCHITZ-SMOOTH FUNCTIONS

**Assumption 11.** *The function $f$ in problem* (1) *is locally Lipschitz-smooth at $x^*$. Specifically, for any $x \in B(x^*, \rho)$, the following inequality holds:*

$$f(x) \leq f^* + \langle \nabla f(x^*), x - x^* \rangle + \frac{L_1}{2}\|x - x^*\|^2, \tag{14}$$

*where $L_1, \rho > 0$ are fixed constants.*

**Lemma 12.** *Assume that $f$ is locally Lipschitz-smooth at $x^*$ with constant $L_1$ (Assumption 11). Then, $\omega(t) \leq \epsilon$ for any given $\epsilon > 0$ whenever $t \leq \delta(\epsilon)$, where*

$$\delta(\epsilon) := \min\left\{ \sqrt{\frac{\epsilon}{L_1}}, \frac{\epsilon}{2\|\nabla f(x^*)\|_*}, \rho \right\}.$$

*Proof.* According to (14), for any $x \in B(x^*, \rho)$, we have

$$f(x) - f^* \leq \|\nabla f(x^*)\|_*\|x - x^*\| + \frac{L_1}{2}\|x - x^*\|^2.$$

Hence, for any $0 \leq t \leq \rho$,

$$\omega(t) \leq \frac{L_1}{2}t^2 + \|\nabla f(x^*)\|_* t.$$

To make the right-hand side $\leq \epsilon$, it suffices to ensure that each of the two terms is $\leq \frac{\epsilon}{2}$:

$$\frac{L_1}{2}t^2 \leq \frac{\epsilon}{2}, \qquad \|\nabla f(x^*)\|_* t \leq \frac{\epsilon}{2}.$$

Solving this system of inequalities, we get the claim. $\qquad\square$

Combining Theorem 1 and Lemma 12, we get the following complexity result.

**Corollary 13.** *Consider problem* (1) *under Assumption 11. Let Algorithm 1 with coefficients* (5) *be applied for solving this problem. Then, $f(x_T^*) - f^* \leq \epsilon$ for any given $\epsilon > 0$ whenever $T \geq T(\epsilon)$, where*

$$T(\epsilon) = \max\left\{ \frac{L_1}{\epsilon}, \frac{4\|\nabla f(x^*)\|_*^2}{\epsilon^2}, \frac{1}{\rho^2} \right\} e^2 D^2 \log^2 \frac{e\bar{D}}{\bar{r}},$$

*and the constants $D$ and $\bar{D}$ are as defined in Theorem 1.*

## D.3 HÖLDER-SMOOTH FUNCTIONS

**Assumption 14.** *The function $f$ in problem* (1) *is locally Hölder-smooth at $x^*$. Specifically, for any $x \in B(x^*, \rho)$, the following inequality holds:*

$$f(x) \leq f^* + \langle \nabla f(x^*), x - x^* \rangle + \frac{H_\nu}{1+\nu}\|x - x^*\|^{1+\nu}, \tag{15}$$

*where $\nu \in [0, 1]$ and $H_\nu, \rho > 0$ are fixed constants.*

**Lemma 15.** *Let $f$ be locally $(\nu, H_\nu)$-Hölder-smooth at $x^*$ (Assumption 14). Then, $\omega(t) \leq \epsilon$ for any given $\epsilon > 0$ whenever $t \leq \delta(\epsilon)$, where*

$$\delta(\epsilon) := \min\left\{ \left[\frac{(1+\nu)\epsilon}{2H_\nu}\right]^{\frac{1}{1+\nu}}, \frac{\epsilon}{2\|\nabla f(x^*)\|_*}, \rho \right\}.$$

*Proof.* According to (15), for any $x \in B(x^*, \rho)$, we have

$$f(x) - f^* \leq \|\nabla f(x^*)\|_*\|x - x^*\| + \frac{H_\nu}{1+\nu}\|x - x^*\|^{1+\nu}.$$

Hence, for any $0 \le t \le \rho$,

$$\omega(t) \le \|\nabla f(x^*)\|_* t + \frac{H_\nu}{1+\nu} t^{1+\nu}.$$

To make the right-hand side of the last inequality $\le \epsilon$, it suffices to ensure that each of the two terms is $\le \frac{\epsilon}{2}$:

$$\|\nabla f(x^*)\|_* t \le \frac{\epsilon}{2}, \qquad \frac{H_\nu}{1+\nu} t^{1+\nu} \le \frac{\epsilon}{2}.$$

Solving this system of inequalities, we get the claim. $\qquad\square$

Combining Theorem 1 and Lemma 15, we get the following complexity result.

**Corollary 16.** *Consider problem* (1) *under Assumption 14. Let Algorithm 1 with coefficients* (5) *be applied for solving this problem. Then,* $f(x_T^*) - f^* \le \epsilon$ *for any given* $\epsilon > 0$ *whenever* $T \ge T(\epsilon)$, *where*

$$T(\epsilon) = \max \left\{ \left[ \frac{2H_\nu}{(1+\nu)\epsilon} \right]^{\frac{2}{1+\nu}}, \frac{4\|\nabla f(x^*)\|_*^2}{\epsilon^2}, \frac{1}{\rho^2} \right\} e^2 D^2 \log^2 \frac{e\bar{D}}{\bar{r}},$$

*and the constants* $D$ *and* $\bar{D}$ *are as defined in Theorem 1.*

### D.4 FUNCTIONS WITH LIPSCHITZ HIGH-ORDER DERIVATIVE

**Assumption 17.** *The function* $f$ *in problem* (1) *is such that its pth derivative is locally* $L_p$-*Lipschitz at* $x^*$. *Specifically,* $f$ *is* $p$ *times differentiable on* $B(x^*, \rho)$, *and, for any* $x \in B(x^*, \rho)$, *the following inequality holds:*

$$\|\nabla^p f(x) - \nabla^p f(x^*)\| \le L_p \|x - x^*\|.$$

*where* $L_p, \rho > 0$ *are fixed constants.*

The Assumption 17 immediately implies the following global upper bound on the function value:

$$f(x) \le f^* + \sum_{i=1}^{p} \frac{1}{i!} \nabla^i f(x^*)[x - x^*]^i + \frac{L_p}{(p+1)!} \|x - x^*\|^{p+1}. \tag{16}$$

**Lemma 18.** *Assume that* $f$ *has locally* $L_p$-*Lipschitz pth derivative at* $x^*$ *(Assumption 17). Then,* $\omega(t) \le \epsilon$ *for any given* $\epsilon > 0$ *whenever* $t \le \delta(\epsilon)$, *where*

$$\delta(\epsilon) := \min \left\{ \min_{2 \le i \le p} \left[ \frac{i! \, \epsilon}{(p+1)\|\nabla^i f(x^*)\|} \right]^{\frac{1}{i}}, \left[ \frac{p! \, \epsilon}{L_p} \right]^{\frac{1}{p+1}}, \frac{\epsilon}{(p+1)\|\nabla f(x^*)\|_*}, \rho \right\}.$$

*Proof.* According to (16), for any $x \in B(x^*, \rho)$, we have

$$f(x) - f^* \le \|\nabla f(x^*)\|_* \|x - x^*\| + \sum_{i=2}^{p} \frac{1}{i!} \|\nabla^i f(x^*)\| \|x - x^*\|^i + \frac{L_p}{(p+1)!} \|x - x^*\|^{p+1}.$$

Therefore, for any $0 \le t \le \rho$, we have

$$\omega(t) \le \|\nabla f(x^*)\|_* t + \sum_{i=2}^{p} \frac{1}{i!} \|\nabla^i f(x^*)\| t^i + \frac{L_p}{(p+1)!} t^{p+1}.$$

To make the right-hand side $\le \epsilon$, it suffices to ensure that each of the following inequalities holds:

$$\|\nabla f(x^*)\|_* t \le \frac{\epsilon}{p+1}, \quad \frac{1}{i!} \|\nabla^i f(x^*)\| t^i \le \frac{\epsilon}{p+1}, \quad \frac{L_p}{(p+1)!} t^{p+1} \le \frac{\epsilon}{p+1}, \quad i = 2, \dots, p.$$

Solving this system of inequalities, we get the claim. $\qquad\square$

Combining Theorem 1 and Lemma 18, we get the following complexity result.

**Corollary 19.** *Consider problem* (1) *under Assumption 17. Let Algorithm 1 with coefficients* (5) *be applied for solving this problem. Then, $f(x_T^*) - f^* \leq \epsilon$ for any given $\epsilon > 0$ whenever $T \geq T(\epsilon)$, where*

$$T(\epsilon) = \max\left\{ \max_{2 \leq i \leq p} \left[ \frac{(p+1)\|\nabla^i f(x^*)\|}{i!\,\epsilon} \right]^{\frac{2}{i}}, \right.$$
$$\left. \left[ \frac{L_p}{p!\,\epsilon} \right]^{\frac{2}{p+1}}, \frac{(p+1)^2 \|\nabla f(x^*)\|_*^2}{\epsilon^2}, \frac{1}{\rho^2} \right\} e^2 D^2 \log^2 \frac{e\bar{D}}{\bar{r}},$$

*and the constants $D$ and $\bar{D}$ are as defined in Theorem 1.*

### D.5 QUASI-SELF-CONCORDANT FUNCTIONS

**Assumption 20.** *The function $f$ in problem* (1) *is Quasi-Self-Concordant (QSC) in a neighborhood of $x^*$. Specifically, it is three times differentiable in a neighborhood of $x^*$ and for any $x \in B(x^*, \rho)$ and arbitrary directions $u, v \in \mathbb{R}^d$, the following inequality holds:*

$$\nabla^3 f(x)[u, u, v] \leq M \langle \nabla^2 f(x)u, u \rangle \|v\|,$$

*where $M \geq 0$ and $\rho > 0$ are fixed constants.*

The following lemma provides an important global upper bound on the function value for QSC functions.

**Lemma 21.** *(Doikov, 2023, Lemma 2.7) Let $f$ be QSC with the parameter $M$. Then, for any $x, y \in \operatorname{dom} f$, the following inequality holds:*

$$f(y) \leq f(x) + \langle \nabla f(x), y - x \rangle + \langle \nabla^2 f(x)(y - x), y - x \rangle \varphi(M \|y - x\|),$$

*where $\varphi(t) := \frac{e^t - t - 1}{t^2}$.*

**Lemma 22.** *Assume that $f$ is a locally QSC function at $x^*$ with constant $M$ (Assumption 20). Then, $\omega(t) \leq \epsilon$ for any given $\epsilon > 0$ whenever $t \leq \delta(\epsilon)$, where*

$$\delta(\epsilon) := \min\left\{ \frac{1}{M}, \sqrt{\frac{\epsilon}{2(e-2)\|\nabla^2 f(x^*)\|}}, \frac{\epsilon}{2\|\nabla f(x^*)\|_*}, \rho \right\}.$$

*Proof.* According to Lemma 21, for any $x \in B(x^*, \rho)$, we have

$$f(x) - f^* \leq \langle \nabla f(x^*), x - x^* \rangle + \langle \nabla^2 f(x^*)(x - x^*), x - x^* \rangle \varphi(M \|x - x^*\|)$$
$$\leq \|\nabla f(x^*)\|_* \|x - x^*\| + \|\nabla^2 f(x^*)\| \|x - x^*\|^2 \varphi(M \|x - x^*\|).$$

Therefore, for any $0 \leq t \leq \rho$, we get

$$\omega(t) \leq \|\nabla f(x^*)\|_* t + \|\nabla^2 f(x^*)\| t^2 \varphi(Mt), \tag{17}$$

where we have used the fact that $\varphi(\cdot)$ is an increasing function.

Note that, for any $0 \leq t \leq \frac{1}{M}$, we can estimate $\varphi(Mt) \leq \varphi(1) = e - 2$. Substituting this bound into (17), we obtain

$$\omega(t) \leq \|\nabla f(x^*)\|_* t + (e - 2)\|\nabla^2 f(x^*)\| t^2.$$

To make the right-hand side $\leq \epsilon$, it suffices to ensure that each of the two terms is $\leq \frac{\epsilon}{2}$:

$$\|\nabla f(x^*)\|_* t \leq \frac{\epsilon}{2}, \qquad (e - 2)\|\nabla^2 f(x^*)\| t^2 \leq \frac{\epsilon}{2}.$$

Solving this system of inequalities, we get the claim. $\qquad \square$

Combining Theorem 1 and Lemma 22, we get the following complexity result.

**Corollary 23.** *Consider problem* (1) *under Assumption 20. Let Algorithm 1 with coefficients* (5) *be applied for solving this problem. Then, $f(x_T^*) - f^* \leq \epsilon$ for any given $\epsilon > 0$ whenever $T \geq T(\epsilon)$, where*

$$T(\epsilon) = \max\left\{ M^2, \frac{2(e-2)\|\nabla^2 f(x^*)\|}{\epsilon}, \frac{4\|\nabla f(x^*)\|_*^2}{\epsilon^2}, \frac{1}{\rho^2} \right\} e^2 D^2 \log^2 \frac{e\bar{D}}{\bar{r}},$$

*and the constants $D$ and $\bar{D}$ are as defined in Theorem 1.*

### D.6 $(L_0, L_1)$-SMOOTH FUNCTIONS

Let us now consider the case when $Q = \mathbb{R}^d$ and $f$ is $(L_0, L_1)$-smooth (Zhang et al., 2020), meaning that for any $x \in \mathbb{R}^d$,

$$\|\nabla^2 f(x)\| \leq L_0 + L_1 \|\nabla f(x)\|_*,$$

where $L_0, L_1 \geq 0$ are fixed constants.

**Lemma 24.** *(Vankov et al., 2024, Lemma 2.2) Let $f$ be $(L_0, L_1)$-smooth. Then, for any $x, y \in \mathbb{R}^d$, it holds that*

$$f(y) \leq f(x) + \langle \nabla f(x), y - x \rangle + \frac{L_0 + L_1 \|\nabla f(x)\|_*}{L_1^2} \xi(L_1 \|y - x\|),$$

*where $\xi(t) := e^t - t - 1$.*

**Lemma 25.** *Assume that $f$ is an $(L_0, L_1)$-smooth function. Then, $\omega(t) \leq \epsilon$ for any given $\epsilon > 0$ whenever $t \leq \delta(\epsilon)$, where*

$$\delta(\epsilon) := \min\left\{ \frac{1}{L_1}, \sqrt{\frac{2\epsilon}{3(L_0 + L_1 \|\nabla f(x^*)\|_*)}}, \frac{\epsilon}{2\|\nabla f(x^*)\|_*} \right\}.$$

*Proof.* According to Lemma 24, for any $x \in \mathbb{R}^d$, we have

$$f(x) - f^* \leq \langle \nabla f(x^*), x - x^* \rangle + \frac{L_0 + L_1 \|\nabla f(x^*)\|_*}{L_1^2} \xi(L_1 \|x - x^*\|)$$

$$\leq \|\nabla f(x^*)\|_* \|x - x^*\| + \frac{L_0 + L_1 \|\nabla f(x^*)\|_*}{L_1^2} \xi(L_1 \|x - x^*\|)$$

Therefore, for any $t \geq 0$, we get

$$\omega(t) \leq \|\nabla f(x^*)\|_* t + \frac{L_0 + L_1 \|\nabla f(x^*)\|_*}{L_1^2} \xi(L_1 t), \tag{18}$$

where the second inequality uses the fact that $\xi(x)$ is an increasing function.

Note that, for any $0 \leq t \leq \frac{1}{L_1}$, we can estimate

$$\xi(L_1 t) \leq \frac{L_1^2 t^2}{2(1 - \frac{L_1 t}{3})} \leq \frac{3}{4} L_1^2 t^2.$$

Substituting this bound into (18), we obtain:

$$\omega(t) \leq \|\nabla f(x^*)\|_* t + \frac{3(L_0 + L_1 \|\nabla f(x^*)\|_*)}{4} t^2.$$

To make the right-hand side of the last inequality $\leq \epsilon$, it suffices to ensure that each of the two terms is $\leq \frac{\epsilon}{2}$:

$$\|\nabla f(x^*)\|_* t \leq \frac{\epsilon}{2}, \qquad \frac{3(L_0 + L_1 \|\nabla f(x^*)\|_*)}{4} t^2 \leq \frac{\epsilon}{2}.$$

Solving this system of inequalities, we get the claim. $\qquad \square$

Combining Theorem 1 and Lemma 25, we get the following complexity result.

**Corollary 26.** *Consider problem* (1) *under the assumption that $f$ is an $(L_0, L_1)$-smooth function. Let Algorithm 1 with coefficients* (5) *be applied for solving this problem. Then, $f(x_T^*) - f^* \leq \epsilon$ for any given $\epsilon > 0$ whenever $T \geq T(\epsilon)$, where*

$$T(\epsilon) = \max\left\{ L_1^2, \frac{3(L_0 + L_1 \|\nabla f(x^*)\|_*)}{2\epsilon}, \frac{4\|\nabla f(x^*)\|_*^2}{\epsilon^2} \right\} e^2 D^2 \log^2 \frac{e\bar{D}}{\bar{r}},$$

*and the constants $D$ and $\bar{D}$ are as defined in Theorem 1.*

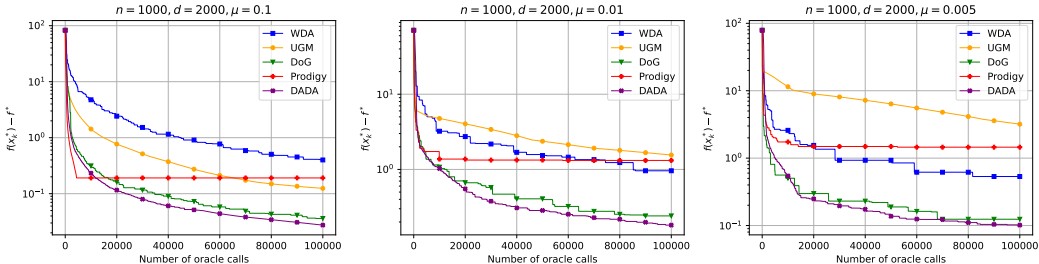

Figure 2: Comparison of different methods on the Softmax function.

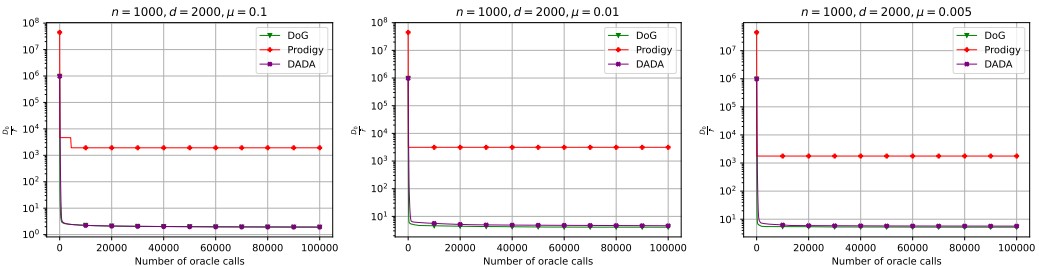

Figure 3: The ratio $\frac{D}{\bar{r}_t}$ for the Softmax function with different optimal points $x^*$.

# E  ADDITIONAL EXPERIMENTS

**Softmax function.**  Our first test problem is

$$\min_{x \in \mathbb{R}^d} \left\{ f(x) := \mu \log \left( \sum_{i=1}^n \exp \left[ \frac{\langle a_i, x \rangle - b_i}{\mu} \right] \right) \right\}, \tag{19}$$

where $a_i \in \mathbb{R}^d$, and $b_i \in \mathbb{R}$ for all $1 \le i \le n$, and $\mu > 0$. This function can be viewed as a smooth approximation of $\max_{1 \le i \le n} [\langle a_i, x \rangle - b_i]$ (Nesterov, 2005a).

To generate the data for our problem, we proceed as follows. First, we generate i.i.d. vectors $\hat{a}_i$ with components uniformly distributed in the interval $[-1, 1]$ for $i = 1, \ldots, n$, and similarly for the scalar values $b_i$. Using this data, we form the preliminary version of our function, $\hat{f}$. We then compute $a_i = \hat{a}_i - \nabla \hat{f}(0)$ and use the obtained $(a_i, b_i)$ to define our function $f$. This way of generating the data ensures that $x^* = 0$ is a solution of our problem.

The results are shown in Fig. 2, where we fix $n = 10^3$ and $d = 2n$, and consider different values of $\mu \in \{0.1, 0.01, 0.005\}$. As we can see, most methods exhibit similar performance for $\mu = 0.1$ except for Prodigy which stops converging after a few initial iterations. This issue, along with a decline in performance for UGM, persists as $\mu$ decreases, whereas DADA, DoG, and WDA remain largely unaffected. Notably, DoG performs very similarly to DADA, which we hypothesize is primarily due to the similarity in estimating $D_0$.

Additionally, Fig. 3 illustrates the ratio between $D_0$ and $\bar{r}$, showing the estimation error of Prodigy, DoG, and DADA throughout the optimization process. For Prodigy, we use $\frac{D_0}{d_{\max}}$ to generate the plot. The figure demonstrates that DADA and DoG exhibit similar behavior in estimating $D_0$, despite employing different update methods—Dual Averaging and Gradient Descent, respectively. However, Prodigy appears to encounter challenges in estimating $D_0$ as its estimation stabilizes at a relatively large value.

**Hölder-smooth function.**  Let us consider the following *polyhedron feasibility problem*:

$$f^* := \min_{x \in \mathbb{R}^d} \left\{ f(x) := \frac{1}{n} \sum_{i=1}^n [\langle a_i, x \rangle - b_i]_+^q \right\}, \tag{20}$$

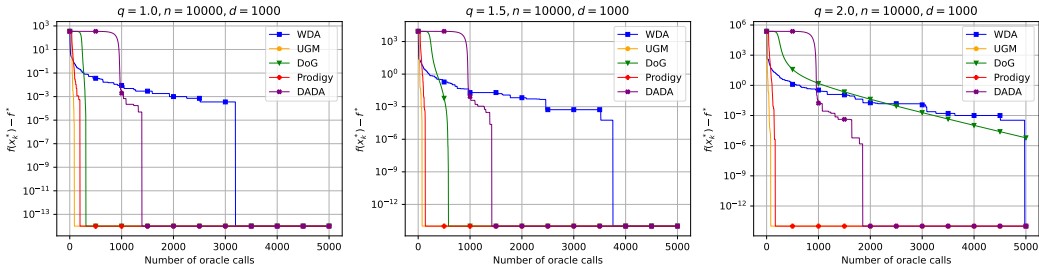

Figure 4: Comparison of different methods on the polyhedron feasibility problem.

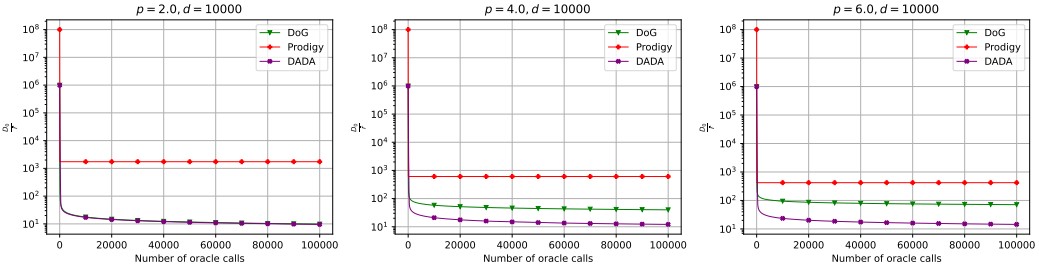

Figure 5: The ratio $\frac{D}{\bar{r}_t}$ for the worst-case function with different optimal points $x^*$.

where $a_i, b_i \in \mathbb{R}^d$, $q \in [1,2]$, and $[\tau]_+ = \max(0, \tau)$. This problem can be interpreted as finding a point $x^* \in \mathbb{R}^d$ lying inside the polyhedron $P = \{x : \langle a_i, x \rangle \leq b_i, \ i = 1, \ldots, n\}$. Such a point exists if and only if $f^* = 0$.

Observe that $f$ in problem (20) is Hölder-smooth with parameter $\nu = q - 1$. Therefore, by varying $q \in [1,2]$, we can check the robustness of different methods to the smoothness level of the objective function.

The data for our problem is generated randomly, following the procedure in (Rodomanov et al., 2024). First, we sample $x^*$ uniformly from the sphere of radius $0.95R$ centered at the origin. Next, we generate i.i.d. vectors $a_i$ with components uniformly distributed in $[-1,1]$. To ensure that $\langle a_n, x^* \rangle < 0$, we invert the sign of $a_n$ if necessary. We then sample positive reals $s_i$ uniformly from $[0, -0.1c_{\min}]$, where $c_{\min} := \min_i \langle a_i, x^* \rangle < 0$, and set $b_i = \langle a_i, x^* \rangle + s_i$. By construction, $x^*$ is a solution to the problem with $f^* = 0$.

We select $n = 10^4$, $d = 10^3$, $R = 10^3$ and consider different values of $q \in \{1, 1.5, 2\}$. As shown in Fig. 4, as $q$ increases and approaches 2, the performance of DoG significantly declines. However, DADA, Prodigy, and UGM demonstrate similar performance regardless of the choice of $q$.

**Worst-case function.** In addition to the experiments presented in Section 4, we evaluate the estimation error of $D_0$ for Prodigy, DoG, and DADA throughout the optimization process, as shown in Fig. 5. The figure illustrates that while Prodigy's estimate of $D_0$, shows some improvement over time, it remains noticeably inaccurate. Moreover, for DoG, the estimate deteriorates as $p$ increases, a behavior that is not observed with DADA, whose estimate remains stable across different values of $p$.

**Comparison of different initial estimates of the distance.** In this experiment, we evaluate the sensitivity of DADA to the choice of the initial point $x_0$. We consider the same Softmax function as in (19) with $n = 10^3$, $d = 2n$, and $\mu \in \{0.5, 0.1, 0.01\}$.

The results are shown in Fig. 6, where we consider $\delta \in \{10^{-1}, \ldots, 10^{-6}\}$. As we can see, the choice of $\delta$ does not affect the performance of DADA, which consistently achieves similar performance across all tested values.

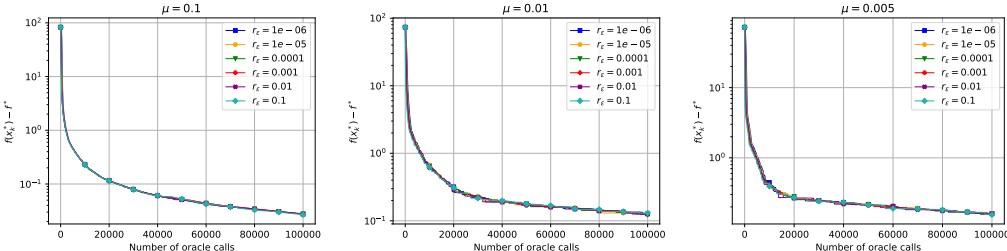

Figure 6: Comparison of different initial estimates of the distance on the Softmax function with different values of $\mu$.

