# OpenReview forum: "DADA: Dual Averaging with Distance Adaptation"
_ICLR.cc/2026/Conference — ICLR 2026 Poster_

### Official Review · Reviewer_UHdw · 2025-10-28

**Soundness:** 3
**Presentation:** 3
**Contribution:** 3
**Rating:** 4
**Confidence:** 4

**Summary:**

This paper introduces a novel universal gradient method for solving convex optimization problems, termed Dual Averaging with Distance Adaptation (DADA). The algorithm is based on the classical dual averaging scheme and dynamically adjusts its coefficients based on observed gradients and the distance between iterates and the starting point. This adaptation eliminates the need for problem-specific parameters, making the method broadly applicable. The authors provide theoretical convergence guarantees and demonstrate the practical effectiveness of DADA through experiments.

**Strengths:**

* The proposed DADA algorithm offers a parameter-free approach to convex optimization, enhancing its applicability across various problems.

*  The paper provides rigorous theoretical convergence guarantees for the proposed method.

* Empirical results demonstrate the practical effectiveness of DADA in solving convex optimization problems.

**Weaknesses:**

* I am a bit concerned about the assumption that $x^* \in \mathrm{int}(dom(f))$ for constrained problems. For example, in a linearly constrained problem where the constraint set $\mathcal{Q}$ is a polyhedron, the optimal solution will often lie on the boundary. It seems that the current work may not fully handle constrained settings, and the results may primarily hold for unconstrained problems (please correct me if I am wrong). I suggest clarifying this or considering an unconstrained version of problem (1).

* I suggest using $g$ rather than $\nabla f(x)$ to denote a subgradient. The notation $\nabla f$ is typically reserved for the gradient of differentiable functions, and using it for subgradients may cause confusion.

* As discussed in the conclusion, it would be interesting to explore whether the proposed methods can be extended to nonconvex and stochastic settings.

* Are the proposed algorithms universal for convex functions satisfying the KL property? Some discussion or analysis would be helpful.

* In Figure 4, could the authors explain why WDA and DADA exhibit sudden decreases after a large number of oracle calls, whereas other methods show smoother decreases? Additionally, in Figure 4, DADA’s performance is not as good as UGM or Prodigy; could the authors provide insight into this behavior?

* Could the authors clarify what convergence rate the proposed algorithm achieves for strongly convex functions?

**Questions:**

Please see above.

---

> ### Author Response · Authors · 2025-11-28
>
> We thank you for your positive feedback on our method’s theoretical and empirical contributions. Below you can find the answers to your questions / comments.
>
> ---
>
> # Questions
>
> > I am a bit concerned about the assumption that $x^* \in \mathrm{int}(\mathrm{dom}(f))$
>  for constrained problems.
>
> Thank you for raising this point. We believe, however, that our assumption $x^\* \in \mathrm{int}(\mathrm{dom}(f))$ does not create any difficulty in the constrained setting. Please note that this assumption does not require the solution $x^\*$ to be an interior point of the feasible set. The feasible set in our problem is $Q$ rather than $\mathrm{dom} f$, and we allow $x^\*$ to be a boundary point of $Q$. Our assumption is simply requiring the function to be defined (= be finite) on a slightly larger set than the feasible set $Q$ and does not contradict your example with $Q$ being polyhedron---think, for instance, about the "classical" situation when $f$ is defined on the entire space, then our assumption $x^\* \in \mathrm{int} (\mathrm{dom} (f))$ is trivially satisfied and $x^\*$ could be any point in $Q$, even at the boundary.
>
> ---
>
> > I suggest using rather than to denote a subgradient. The notation is typically reserved for the gradient of differentiable functions, and using it for subgradients may cause confusion.
>
> Thank you for your comment. As in the previous section, we intentionally used the notation $\nabla f(x)$ to maintain simplicity in the main text. This choice helps general readers who may not be familiar with subgradients to follow the text, while still covering nondifferentiable objective functions. However, we understand the potential confusion, and we will add a remark clarifying this choice in the revised version.
>
> ---
>
> > In Figure 4, could the authors explain why WDA and DADA exhibit sudden decreases after a large number of oracle calls, whereas other methods show smoother decreases? Additionally, in Figure 4, DADA’s performance is not as good as UGM or Prodigy; could the authors provide insight into this behavior?
>
> We believe that this behavior is mainly due to the specific problem class used in Figure 4. As explained in the paper, the objective function is constructed from the constraints defining the polyhedron
>
> $$
> P = \{x : \langle a_i, x \rangle \leq b_i,\ i = 1,2,\ldots,n\}.
> $$
>
> Concretely, the objective measures the violation of these linear constraints, so it is (essentially) zero for $x \in P$ and positive outside $P$. This means that if, at some iteration, the algorithm takes a step that moves the iterate from outside $P$ to inside $P$, the objective value can drop sharply in a single iteration. This explains the sudden decreases observed for WDA and DADA: both methods can keep relatively large effective step sizes while they are still outside the feasible region, and once they cross the boundary of $P$ the violation penalty vanishes, producing a steep drop in the plot. Methods with smoother steps tend to approach the boundary more gradually and therefore produce smoother curves.
>
> Regarding the second point (DADA vs. UGM and Prodigy), this particular problem is very favorable to methods that maintain aggressive steps near the boundary of $P$: once the feasible region is reached, there is no need to refine the solution further, since the objective is already at its minimal value. UGM and Prodigy behave in this way on this instance, entering $P$ earlier and thus reaching low objective values with fewer oracle calls. DADA, by design, is more conservative in order to be robust across a wide range of growth regimes: its step-size adaptation keeps a bit more regularization near the boundary, so it spends more iterations close to $P$ before actually entering it. This leads to slightly worse performance on this particular feasibility-type problem, even though DADA still eventually attains the same (zero-violation) objective value.
>
> ---

---

> > ### Author Response · Authors · 2025-11-28
> >
> > > Could the authors clarify what convergence rate the proposed algorithm achieves for strongly convex functions?
> >
> > Thank you for this question. Since the KL property can be viewed as a generalization of strong convexity, we address both aspects together here. Similarly to the classical subgradient method or SGD, our algorithm does not automatically accelerate on strongly convex functions. This is a consequence of the distance-adaptive stepsize design, which is driven by the initial distance $D_0 = ||x_0 - x^*||$. This design choice is what allows us to obtain a single set of guarantees that simultaneously covers many different function classes.
> >
> > To obtain better rates on strongly convex problems, we need to further modify our method to take advantage of strong convexity. However, it is not clear how to do this without some knowledge of the problem class or its parameters, which we deliberately avoid in this work to preserve universality. Indeed, one natural idea could be to periodically restart the method to exploit the fact that $x_k^\* \to x^\*$. Specifically, we could restart when the function residual halves, that is when $f_k \equiv f(x_k^\*) - f^\* \leq \frac{1}{2} f_0$, where $f_0 \equiv f(x_0) - f^\*$ is the initial residual. In this form, the strategy may not be implementable, since $f^\*$ is typically unknown. Therefore, one would first apply our convergence guarantee $f_k \leq \omega(\frac{D_0}{\sqrt{k}})$ (in a simplified form, omitting the logarithmic factor inside), where $D_0 = || x_0 - x^\* ||$ is the initial distance, and then use strong convexity to relate $\frac{\mu}{2} D_0^2 \leq f_0$. This gives us $f_k \leq \omega(\frac{2 f_0}{\mu k})$, and we should restart whenever the right-hand side is $\leq \frac{1}{2} f_0$. This, however, requires knowing the growth function $\omega$, i.e., a particular problem class and its parameters.
> >
> > Alternatively, instead of using restarts, one could attempt to modify the current convergence analysis and strengthen some of the convexity inequalities by adding the quadratic term coming from strong convexity. There is, however, only one place in the current analysis where convexity is used—inequality (3), which connects the function residual $f(x_k) - f^*$ with the hyperplane distance $v_k$. While it is possible to strengthen this inequality for strongly convex functions, the resulting bound does not seem to help much, at least without knowledge of the growth function $\omega$.

---

### Official Review · Reviewer_xex2 · 2025-10-31

**Soundness:** 4
**Presentation:** 3
**Contribution:** 3
**Rating:** 6
**Confidence:** 3

**Summary:**

The paper studies gradient methods that can adapt to the distance between the initial solution and the optimal solution. Previous works have developed such algorithms for various settings, both deterministic and stochastic with different assumptions on the gradients. This work shows a single algorithm that works simultaneously for several classes of functions including Lipschitz functions, Lipschitz-smooth functions, Holder-smooth functions, functions with higher order Lipschitz derivative, quasi-self-concordant functions, and (L0, L1) smooth functions. The algorithm is based on the dual averaging method combined with a recent technique for adapting to the distance to the optimal solution, previously applied to gradient descent algorithms.

**Strengths:**

The new method works simultaneously for many classes of functions, for both constrained and unconstrained problems. The proof is done via a unified approach and the bounds for each function class is a relatively simple corollary of the main proof.

**Weaknesses:**

The new method does not work for the stochastic case whereas some previous works include the stochastic case.

There is already a previous method that is universal for several classes of functions listed but that work did not consider others like Lipschitz higher order derivative.

In theory, one can use classical methods like the doubling trick to adapt to the distance and can combine even with tools like acceleration. Thus, the main benefit here is that one does not need to restart.

**Questions:**

Could you please comment on the barrier for obtaining a method with acceleration?

The previous work Orabona. https://arxiv.org/pdf/2308.05621 seems to suggest that one can combine normalized gradient descent with different algorithms in a similar fashion. The previous "universal" result by Khaled et al. used gradient descent whereas this paper uses DA. Is there a fundamental reason for this or do you see your technique being applicable to gradent descent as well?

---

> ### Author Response · Authors · 2025-11-28
>
> Thanks for your time and effort spent on reviewing this manuscript. We appreciate the feedback and would like to make some comments on the points you raised in your review.
>
> We agree that extending to stochastic settings is important, but we view building a solid, adaptive deterministic framework as a necessary first step. We also acknowledge prior universal methods; our notion of universality covers a broader set of convex classes while remaining fully adaptive. We likewise agree that techniques such as the doubling trick or explicit restarts can be used to obtain adaptive methods. In designing our method, an important goal was to preserve conceptual simplicity and practical efficiency by achieving adaptivity without relying on these mechanisms.
>
> ---
>
> # Questions
>
> > Could you please comment on the barrier for obtaining a method with acceleration?
>
> Exploring the possibility of providing an accelerated version of DADA is indeed a very interesting and important direction, which we leave for future research.
>
> Let us provide some intuition on why this is a difficult problem to tackle. Since acceleration is hardly tied to a single problem class, it is not even fully clear in which terms to write down a universally meaningful convergence estimate. Concretely, our analysis is separated into two components:
>
> $$f(x_T^\*) - f^\* \leq \omega(v_T^\*),$$
>
> where
> $$
> v_T^* = \tilde{\mathcal{O}}(\frac{D}{\sqrt{T}}).
> $$
> The first bound holds for any sequence of points and depends on the specific function class through the growth function $\omega(\cdot)$, while the second bound is class-agnostic and is instead a property of our concrete method. A natural hope would be to accelerate the class-agnostic part, improving $v_T^* = \widetilde{\mathcal{O}}(D_0 / \sqrt{T})$ to something faster such as $v_T^* = \widetilde{\mathcal{O}}(D_0 / T)$. However, this turns out to be impossible in a universal sense.
>
> Indeed, if one could establish a bound of the form $v_T^* \lesssim 1/T$, then for $L_1$-smooth functions, where $\omega(t) \lesssim L_1 t^2$, inequality (3) would immediately yield an accelerated rate
> $$
> f(x_T^\*) - f^\* \lesssim \frac{L_1}{T^2}.
> $$
> However, this type of universal improvement is incompatible with the optimal complexity for nonsmooth Lipschitz functions. When $f$ is $L_0$-Lipschitz, the growth function satisfies $\omega(t) \le L_0 t$, and (3) implies
> $$
> f(x_k) - f^* \le L_0 v(x_k)
> \quad\text{for every iterate } k.
> $$
> Any algorithm guaranteeing $v_T^* = O(1/T)$ uniformly over this class would therefore satisfy
> $$
> f(x_T^\*) - f^\* = O\left(\frac{L_0 D_0}{T}\right),
> $$
> in contradiction with the classical lower bound $\Omega(L_0^2 D_0^2 / \varepsilon^2)$ for nonsmooth convex optimization with a first-order oracle. In this sense, the $O(1/\sqrt{T})$ behavior of $v_T^*$ in Theorem. 1 is already of the correct order for the worst-case Lipschitz class, and there is no room to make the universal bound faster.
>
> ---
>
> > Is there a fundamental reason for this or do you see your technique being applicable to gradent descent as well?
>
> Thank you for pointing this out. There is no fundamental barrier to extending our analysis to the (projected) gradient–descent / subgradient scheme. The only drawback is that, in the GD formulation, the step sizes that give the same guarantees as our dual–averaging scheme are of the ``predefined'' type and depend on the total number of iterations $T$. If $T$ is not known in advance, one can still recover essentially the same guarantees at the price of an additional logarithmic factor in the rate.
>
> At a high level, the only place where we use the dual–averaging scheme is Lemma 5, where we obtain
>
> $$
> \sum_{i = 0}^{k - 1} a_i v_i ||g_i||_{\*} + \frac{\beta_k}{2} D_k^2 \leq \frac{\beta_k}{2} D_0^2 + \sum\_{i = 0}^{k - 1} \frac{a\_i^2}{2\beta\_i} ||g\_i||\_{\*}^2.
> $$
>
> with $D_k = ||x_k - x^*||$. For a GD–type algorithm, it turns out that the same recursion can be established in the special case $\beta_i = 1$, which yields
>
> $$
> \sum_{i = 0}^{k - 1} h_i v_i \leq D_0^2 + \frac{1}{2} \sum_{k = 0}^{k - 1} h_i^2,
> $$
>
> where $h_k$ is our step-size at step $k$. We could then select $h_k \simeq \frac{\bar{r}_k}{\sqrt{T}}$, using the same quantity $\bar{r}_k$ as in the paper, with explicit dependence on the total number of iterations $T$. This would give us the same convergence rate for this method after $T$ iterations. However, if we do not want to fix $T$ in advance, we need to use a more cautious step-size sequence, involving something like $\sqrt{k} \log k$ instead of $T$, which results in both a worse theoretical bound and worse practical performance. A similar situtation occurs for DoG, where the method can potentially diverge without the additional $\log k$ factor in the step-size. For further discussion, see the section “DoG can run wild” in [IHC23].

---

> > ### Author Response · Authors · 2025-11-28
> >
> > Finally, plugging this inequality in place of Lemma 5, one can reproduce our convergence guarantees for a GD–type algorithm. We therefore view the restriction to the dual–averaging scheme as mainly a matter of presentation and not as a fundamental limitation of the approach.
> >
> > ---
> >
> > ## References
> >
> > - [IHC23] M. Ivgi, O. Hinder, Y. Carmon. DoG is SGD's Best Friend: A Parameter-Free Dynamic Step Size Schedule. ICML, 2023.

---

### Official Review · Reviewer_7W2a · 2025-11-01

**Soundness:** 3
**Presentation:** 3
**Contribution:** 3
**Rating:** 2
**Confidence:** 3

**Summary:**

This paper proposes DADA, a parameter-free version of dual averaging (DA). The main idea of parameter-free methods is to adaptively tune the step sizes based on the observed distance of iterates from the starting point to reduce the cost incurred by the (estimated) initialization scale (denoted by $\rho$ in the paper). Applying this to DA, we can adjust the parameters based on a running estimate of an adaptively estimated lower bound of $D_t$, and the authors show that this yields convergence guarantees having a smaller $\log^2 \rho$ dependence while maintaining rates with respect to other factors. The paper also shows theoretical results under applications to a wide variety of function classes (nonsmooth Lipschitz, Lipschitz-smooth, Hölder-smooth, functions with Lipschitz higher-order derivatives, quasi-self-concordant, and $(L_0, L_1)$-smooth). The rates usually match previously known methods up to logarithmic factors while maintaining the benefits of distance adaptation.

**Strengths:**

- It seems like a good idea to use an adaptive, increasing estimate of $D$’s, with a similar spirit (but different in detail) as in methods like D-adaptation, to reduce the cost incurred by estimated initialization scale to a logarithmic term. This allows the algorithm to gain similar benefits in (almost) all the broader classes of convex problems that classical dual averaging can deal with.
- The theoretical statements, proof sketches, applications to examples, and comparison with previous work are all clear. There also are empirical results that align with the theory proposed in the paper.

**Weaknesses:**

- I am currently giving a rating of 2 because **the format of the submission (exceeding 9 pages) falls into desk rejection criteria.** (*I will update my score if the paper somehow avoids desk rejection.*)
- See Questions for other details.

**Questions:**

- Are the guarantees for applications should be essentially thought of as considering unconstrained problems? The problem and the main theorem claims to focus an objective with a constrained form, but we have to assume that projections are be easy to compute in order to seamlessly translate iteration complexity into the actual computational costs, and for the applications we also need assumptions like $\nabla f(x^{\star}) = 0$ to do the theories, both of which makes me feel like it’s a slight overclaim to advocate DADA as an algorithm with clear theoretical guarantees for constrained problems. Can the authors provide what can we say about DADA on constrained cases, or are there experimental results on constrained problems that can show consistency of the analyses?
    - The updates of dual averaging type algorithms have some structural similarities to FTRL updates of the form [1, 3]
    $$ x_{t+1} = \operatorname{argmin}\_{x \in \mathcal{X}} \left\\{ \sum_{i=0}^{k-1} \langle g_i, x_i \rangle + \frac{1}{2 \eta_k} \\| x - \textcolor{red}{x_0} \\|^2  \right\\} $$
    or Frank-Wolfe updates of the form [2, 3]
    $$ \begin{align*}
    d_k &= \sum_{i=0}^{k-1} g_i + \frac{1}{2 \eta_k} (x - \textcolor{red}{x_0}) \\\\
    v_k &= \operatorname{argmin}\_{v \in \mathcal{X}} \\langle d_k, v \\rangle \\\\
    x_{k+1} &= (1 - \sigma_k) x_k + \sigma_k v_k
    \end{align*} $$
    which are both good ways to solve *constrained* optimization problems (FW being *projection free*). In particular, [3] considers updates (with the ${\textcolor{red}{x_0}}$'s) that look exactly as above but uses $\eta_k \equiv \eta$ and $\sigma_k \equiv \sigma$ for technical reasons (in order to use automated proof generation via PEP).
    - I am not very familiar with the proposed (classical) dual-averaging algorithms in the paper; could there be a possibility that using distance adaptation and appropriately scaled regularizers (ex. try to match the $\alpha_k$ and $\beta_k$ of dual averaging and DADA) somehow yield regret/FW-gap upper bounds with better dependence on initialization scale, or this doesn't make any sense because of the $\alpha_k$ and $\beta_k$'s or for some other reason? (This is a light question, I only expect a simple high-level answer on the authors' take on this.)
- Have the authors tried experiments on the stochastic setting as well? Up to my knowledge, methods like D-adaptation also work quite well under stochastic settings, despite the theoretical statements holding only for deterministic oracles, and I expect something similar for DADA as well.

[1] Hazan, E. and Kale, S. 2012. Projection-free online learning. In Proceedings of the 29th International Coference on International Conference on Machine Learning (ICML'12).

[2] Orabona, F. (2019). A Modern Introduction to online learning. ArXiv, abs/1912.13213.

[3] Weibel, J., Gaillard, P., Koolen, W. M., and Taylor, A. Optimized projection-free algorithms for online learning: construction and worst-case analysis. 2025. ⟨hal-05097004⟩

---

> ### Author Response · Authors · 2025-11-28
>
> Thank you for your time and effort spent on reviewing our manuscript. We appreciate your feedback and would like to respond to the points you raised in your review.
>
> We sincerely apologize for the mistake with the formatting: due to an oversight, we exceeded the 9-page limit by three lines. Despite this issue, we still hope and would appreciate having a meaningful discussion regarding the contents and contributions of our paper.
>
> ---
>
> # Questions
>
> > Relation with FTRL and Frank-Wolfe.
>
> Dual averaging is indeed a fairly general framework with two sequences of coefficients $a_k$ and $b_k$. If one of these sequences is kept constant, e.g., $a_k = 1$, the update coincides with the standard FTRL update with a quadratic regularizer. In that sense, FTRL can be viewed as a special case of (classical) dual averaging with uniform weights. We expect that our distance-adaptation idea could also be applied in this FTRL setting to remove the need for a good a priori estimate of $D = ||x_0 - x^*||$. However, the universality guarantees we prove for DADA rely on the specific weighting and normalization we use, so a distance-adaptive FTRL variant would likely require a different analysis and may not achieve the same breadth of guarantees.
>
> The situation for Frank–Wolfe is rather different. FW-type methods are designed for bounded feasible sets (otherwise, the update $v_k$ is not even well defined), and their rates depend on the diameter of the domain rather than on the distance from the initial point to the solution. Moreover, standard FW algorithms are essentially parameter-free and typically do not require tuning problem-dependent constants. On the other hand, they usually exhibit slower convergence than classical gradient methods (for example, they do not necessarily converge on general nonsmooth Lipschitz functions). For these reasons, distance adaptation in the sense we use it does not seem particularly natural or beneficial for the classical FW setting.
>
> ---
>
> > Are the guarantees for applications should be essentially thought of as considering unconstrained problems?
>
> Thank you for mentioning this. Please note that our guarantees are not restricted to unconstrained problems. The algorithm and Theorem 1 are stated for the general constrained problem (1) with $x \in Q$. We assume the set $Q$ is sufficiently simple so that it admits an efficient Euclidean projection. This is a very classical assumption, satisfied by many basic yet important constraint sets in applications, such as the Euclidean ball, the nonnegative orthant, box constraints, or low-dimensional linear subspaces.
>
> In Section 3, we specialize the presentation to the case $\nabla f(x^*) = 0$, which in particular covers unconstrained problems. We do this only to keep the discussion of the different function classes as clean and readable as possible, and also to reflect an important practical setting. The method itself does **not** require this assumption for convergence, and the corresponding rates for the general constrained case are derived in Appendix C. In the revised version, we will add a short remark at the beginning of Section 3 explicitly stating that the results in Appendix C cover the general constrained setting as well.

---

### Author Response · Authors · 2025-11-28

We first thank all reviewers for their time, careful reading, and many insightful comments and questions about our work. We have done our best to answer each point honestly and in detail in the individual responses below. However, several critiques and questions recur across multiple reviews, and we would therefore like to write one general comment addressing these common concerns.

Several reviewers see as a major limitation of our submission the fact that we currently focus on the “basic” setting of deterministic convex optimization and do not yet include extensions to stochastic optimization, strongly convex problems, accelerated methods, and other important settings. We fully agree that all these directions are essential, and we see them as natural follow-up projects. At the same time, we believe that, once we insist on keeping the method as universal as it currently is—one algorithm that works without changes for many different classes of convex problems—the corresponding extensions become technically quite nontrivial, and in some cases may even be impossible without sacrificing universality. This is because the construction of stochastic, strongly convex, or accelerated methods is usually designed for a specific class (e.g., Lipschitz-smooth functions) and relies heavily on its particular structure. It is not clear how to adapt them while still having one single algorithm that provides “out of the box” guarantees for all the function classes we consider. In our per-reviewer replies, we provide more intuition and concrete examples of the obstacles for each requested extension. Even when such extensions are possible, they would deserve a dedicated paper to preserve the clarity and simplicity of the present work.

In return for this narrower scope, we would like to emphasize that even in the current deterministic convex setting, our contributions are already quite substantial and nontrivial.

1. We design a rather simple dual-averaging-type algorithm, DADA, that provides a broader notion of “universality” than classical universal gradient methods such as Nesterov’s UGM [N15]. In particular, our method provides convergence guarantees for high-order smooth, quasi-self-concordant, and $(L_0,L_1)$-smooth functions. Existing universal gradient methods do not cover these classes.
2. For some specific problem classes, such as functions with Lipschitz high-order derivatives and quasi-self-concordant functions, we are not aware of any other first-order methods that match the complexity guarantees obtained by DADA (see, e.g., the discussion in Section 3). In these cases, our analysis appears to be the first to establish such convergence rates, without requiring knowledge of any problem-dependent parameters.
3. We are not aware of any competing algorithm that is as universal as DADA in the sense of providing nontrivial complexity bounds for every subclass in the family of convex function classes we consider, while also matching our guarantees on those classes. For instance, we discuss in the paper that DoG also appears “as universal,” but its complexity results are, in fact, worse than ours.

For these reasons, we believe that, even without the additional extensions mentioned above, the present work already makes a substantial contribution. Therefore, we kindly ask the reviewers to take these points into account when reassessing the importance of our results and to consider raising their scores. We would be happy to further clarify any aspect of the method, theory, or comparisons that remains unclear.

---

References:

- [N15] Y. Nesterov. Universal gradient methods for convex optimization problems. Math. Program, 2015.

---

### Meta-Review · Area_Chair_YnEz · 2025-12-17

**Summary:**

The idea that it is possible to have a method that automatically adapts to different smoothness assumptions is very nice. I think that both the reviewers and I agree with that. Based on only this, the paper should be accepted.

*Weaknesses that should be fixed in the camera-ready paper:* i) I should remind the authors that the paper violates the ICLR paper format (e.g., margins are incorrect), so don't forget to fix this; ii) the paper claims that it can work with constraints problem, which seems to be true. However, in Appendix C, when $\nabla f(x^*) \neq 0$ the convergence rates are much worse with the dependence $\frac{1}{\varepsilon^2}$ on $\varepsilon$ (if I am not missing something). **This should be highlighted in the main part of the paper, at the beginning of Section 3.** Right now, it is hidden in the appendix.; iii) please also incorporate other usefull comments of your rebuttal to the camera-ready paper.

**Reviewer Concerns:**

I think all reviewers' concerns are addressed. The only major non-addressed concern is the lack of a stochastic setting, which I believe is not a major weakness for the line of work that considers adaptive methods.

**Reviewer Scores:**

I think Reviewer 7W2a and Reviewer UHdw would increase the scores since the authors addressed their comments and questions. Moreover, PCs decided not to desk reject the paper since the paper has narrower horizontal margins but wider margins on the bottom. Considering this decision, Reviewer 7W2a would probably increase the score.

---

### Decision · Program_Chairs · 2026-01-26

Accept (Poster)